# An analysis of the interaction between surface and basal crevasses in ice shelves

Maryam Zarrinderakht[1], Christian Schoof[1], and Anthony Peirce[2]

[1]Department of Earth, Ocean and Atmospheric Sciences, University of British Columbia, BC, Canada
[2]Department of Mathematics, University of British Columbia, BC, Canada

**Correspondence:** M. Zarrinderakht (mzaryam@eoas.ubc.ca)

**Abstract.** The prescription of a simple and robust parameterization for calving is one of the most significant open problems in ice sheet modelling. One common approach to modelling of crevasse propagation in calving in ice shelves has been to view crevasse growth as an example of linear elastic fracture mechanics. Prior work has however focused on highly idealized crack geometries, with a single fracture incised into a parallel-sided slab of ice. In this paper, we study how fractures growing from opposite sides of such an ice slab interact with each other, focusing on different simple crack arrangements: we consider either perfectly aligned cracks, or periodic arrays of laterally offset cracks. We visualize the dynamics of crack growth using simple tools from dynamical systems theory, and find that aligned cracks tend to impede each other's growth due to the torques generated by normal stresses on the crack faces, while periodically offset facilitate simultaneous growth of bottom and top cracks. For periodic cracks, the presence of multiple cracks on one side of the ice slab however also generates torques that slow crack growth, with widely spaced cracks favouring calving at lower extensional stresses than closely spaced cracks.

## 1 Introduction

Iceberg calving is a key process in the dynamics of marine ice sheets, since it regulates the length of floating ice shelves and consequently the rate at which ice is discharged across the grounding line (Schoof et al., 2017; Haseloff and Sergienko, 2018). Although there are numerous distinct approaches to model calving (e.g. Bassis and Ma, 2015; Benn et al., 2017; Cook et al., 2014; Fastook and Schmidt, 1982; Nick et al., 2010, 2013; Nye, 1955; Todd and Christoffersen, 2014), there is currently no widely-accepted "universal calving law" that can be applied in large scale ice sheet and glacier flow models.

Calving generally happens as the result of cracks growing to occupy the full thickness and width of an ice shelf. One common approach to model such cracks is to consider ice as elastic medium on the short time scales associated with fracture propagation, and to employ either a classical linear elastic fracture mechanics approach (Weertman, 1973, 1980; van der Veen, 1998a,b; Lai et al., 2020; Zarrinderakht et al., 2022), or to treat crack propagation as the result of breaking discrete bonds (Bassis, 2011; Åström et al., 2013; Crawford et al., 2021). The former leads to continuum models that identify the strength of stress singularities at crack tips as controlling whether a crack will propagate or not (Zehnder, 2012), and the computation of these "stress intensity factors" can be quite elaborate. In practice, this has restricted the use of linear elastic fracture to simple ice geometries with a single crack (Weertman, 1973, 1980; van der Veen, 1998a,b; Lai et al., 2020; Zarrinderakht et al., 2022),

for which interpolated Green's functions can be used to predict fracture propagation (Tada et al., 2000). This approach allows simple calving laws to be derived (Lai et al., 2020; Zarrinderakht et al., 2022) but leaves many open questions, such as whether crevasses incised into the bottom and top of the ice will typically grow simultaneously and meet in the middle to cause calving (as is implicitly assumed in some simpler calving models (e.g. Nick et al., 2010)), whether there is a preferred spacing between crevasses, and how the presence of multiple crevasses affects the calving laws constructed in Lai et al. (2020) and Zarrinderakht et al. (2022).

Discrete element models (Bassis, 2011; Åström et al., 2013; Crawford et al., 2021) are better able to cope with multiple interacting cracks, and with cracks of arbitrary geometry, but they are computationally expensive and therefore difficult to apply when exploring larger regions of parameter space. More recently, phase-field models for fracture mechanics have been applied to crevasse formation (e.g. Clayton et al., 2022; Sondershaus et al., 2023), which reproduce the predictions of linear elastic fracture mechanics closely while also being able to handle phenomena such as crack splitting and viscoelastic relaxation of stresses (though, at present, seemingly only for small viscous strains). As with discrete elements, phase field models however are also computationally more expensive than classical linear elastic fracture mechanics approaches, requiring additional degrees of freedom to be solved for. Note that more general damage mechanics models (Duddu and Waisman, 2013b,a; Duddu et al., 2020; Jimeénez et al., 2017; Keller and Hutter, 2014; Mobasher et al., 2016) aim in a similar direction, but unlike phase field models are not ostensibly based on the energetics of creating new fracture surfaces, and introduces additional parameters that control not only a critical stress for damage production, but also the rate of damage production, which makes comparison with models based on fracture mechanics more difficult.

Here we attempt to bridge the gap between idealized classical fracture mechanics models and more complicated (and computationally much more expensive) discrete element as well as phase field models by extending prior work on linear elastic fracture mechanics models to take account of multiple interacting cracks. We use the boundary element method described in Zarrinderakht et al. (2022), which can in principle handle arbitrary domain and crack geometries, to solve for stress intensity factors at crack tips and solve for their propagation.

To keep the scope of our work tractable, we restrict ourselves to understanding simple interactions between basal and surface crevasses. In particular, we seek to identify how the spacing and alignment of crevasses on opposite sides of an ice shelf affect calving. Note that the study of interacting cracks has a long history, often involving complicated geometries in which the direction of crack propagation must be determined as part of the solution of the linear elastic fracture mechanics problem (e.g. Segall and Pollard, 1980; Baud and Reuschlé, 1997). Here we use the fact that two-dimensional ice shelves flow in pure shear at leading order (Morland, 1987) to restrict ourselves to simple crack geometries in which the stress field remains symmetric about each crack and the assumption of vertical crack propagation remains self-consistent with a maximum hoop stress criterion (Zehnder, 2012). Importantly, this does not imply that spontaneous symmetry breaking cannot occur in the geometry considered (for instance, if the direction of crack propagation is determined by a maximum energy release criterion, see also Zehnder (2012)); our formulation simply does not consider this possibility;

The paper is structured as follows: in section 2, we sketch the model presented by Zarrinderakht et al. (2022) and extend it to accommodate the concurrent growth of multiple cracks. We show that the simultaneous propagation of two cracks can be

cast as a two-dimensional semi-smooth dynamical system in terms of the two crack lengths and a minimal set of dimensionless parameters. We adopt the same weakly inertial crack propagation criterion used in Zarrinderakht et al. (2022). We assume that the total stress field is composed of an elastic stress induced by the newly introduced cracks, and a viscous pre-stress, and impose contact constraints that prevent opposite crack faces from penetrating into each other. In section 3.1, we describe how crack propagation and the eventual steady state configuration that is attained from given initial conditions can be visualized using a phase plane. In section 3.2, we explore how changes in parameters affect those steady state configurations and ultimately leads to calving. Having focus initially on two aligned cracks in a wide domain, we find that such cracks naturally inhibit each other's growth. We therefore explore a periodic configuration with laterally offset cracks in section 3.3, finding qualitatively different behaviour in which calving due to growth of both crevasses is more prevalent. The results obtained from a periodic domain are however sensitive to domain width, and we explore the effect of lateral crack spacing in section 3.4. We discuss implications of our results in section 3.5.

## 2  Model

The basic model is described in Zarrinderakht et al. (2022). In a Cartesian coordinate system $(x_1, x_2) = (x, z)$ where $z = 0$ is at sea level, we consider a parallel-sided slab of ice between $z = s$ and $z = b$, $s$ being surface elevation and $b$ basal elevation. The ice slab is assumed to be part of a larger ice shelf afloat in the ocean, so $s = (1 - \rho_i/\rho_w)H$ and $b = -(\rho_i/\rho_w)H$, where $H$ is the thickness of the ice, and $\rho_i$ and $\rho_w$ are the densities of ice and water, respectively. We model only the portion of the slab of ice between $x = 0$ and $x = W$. The ice is subject to a viscous pre-stress of the form

$$\sigma_{11}^v = \rho_i g(s - z) + R_{xx}, \qquad \sigma_{22}^v = \rho_i g(s - z), \qquad \sigma_{12}^v = \sigma_{21}^v = 0, \tag{1}$$

where $g$ is the acceleration due to gravity, and $R_{xx}$ is related to the far-field velocity field $U$ through $R_{xx} = 4\mu \partial U/\partial x$, $\mu$ being ice viscosity (van der Veen, 1983; Muszynski and Birchfield, 1987; Morland, 1987; MacAyeal and Barcilon, 1988). We consider the short-term elastic response (on time scales much less than a single Maxwell time) to the introduction of cracks into the lower and upper boundaries of the slab of ice. That elastic response takes the form of elastic stress that is added to the viscous pre-stress, and the high stress concentrations at the tip of the cracks can cause the cracks to propagate.

As in Zarrinderakht et al. (2022), we prescribe a fluid pressure on the domain boundary: any part of the lower boundary below sea level is subject to hydrostatic water pressure in the ocean, while at the upper boundary, we prescribe a surface water level at

$$z = s - h_w. \tag{2}$$

Any part of the upper surface below that elevation is also subject to a hydrostatically increasing water pressure, with the water level remaining unchanged as surface cracks propagate (see also Figure 1). Implicit here is the presence of a near-surface aquifer that can supply sufficient water to fill the crack while maintaining that constant water level. As in Zarrinderakht et al. (2022), we ignore the effect of elastic displacements on the fluid pressure at the boundary, thereby omitting buoyancy effects.

This is a potentially significant omission that affects large-scale flexure effects discussed further in section 4.2 below (see also sections 2.1 and 6.3 of Zarrinderakht et al. (2022)).

We apply the same contact-type boundary conditions on crack faces as described in Zarrinderakht et al. (2022), meaning that normal compressive stress either attains the prescribed water (or atmospheric) pressure and the crack faces have moved apart, or normal compressive stress is at or above that fluid pressure, and the crack faces are touching. Other parts of the surface simply experience normal stress equal to the external fluid pressure, while shear stresses are assumed to vanish everywhere on the external boundary.

At the lateral domain boundaries at $x = 0$ and $x = W$, we apply either the same conditions of vanishing elastic shear and normal stress as in Zarrinderakht et al. (2022), or we apply periodic boundary conditions on elastic displacement and stress: in the notation of Zarrinderakht et al. (2022), the latter corresponds to

$$u_i(0, z, t) = u_i(W, z, t), \qquad \text{and} \qquad \sigma_{i1}^{\mathrm{e}}(0, z, t) = \sigma_{i1}^{\mathrm{e}}(W, z, t), \tag{3}$$

where $u_i$ is the displacement field, $\sigma_{ij}^{\mathrm{e}}$ is the elastic stress as defined in Zarrinderakht et al. (2022), and $i$ runs over $\{1, 2\}$.

We have previously considered only a single surface or basal crack in Zarrinderakht et al. (2022), in line with van der Veen (1998a,b) and Lai et al. (2020). Our goal in the present paper is to understand better how cracks interact with each other, focusing on the interaction between basal and surface cracks. To generalize our previous work but still retain enough simplicity to allow for qualitative insight, we consider one basal and one surface crack in the domain, of lengths $d_{\mathrm{t}}$ and $d_{\mathrm{b}}$, respectively, and assume that both are oriented vertically. The symmetry conditions we impose on their locations below makes that choice of orientation self-consistent.

The elastostatic problem of Zarrinderakht et al. (2022) allows us to compute a stress intensity factor $K_{\mathrm{It}}$ and $K_{\mathrm{Ib}}$ at the tip of the surface and basal cracks, respectively, given the current domain geometry as well as material properties and forcing parameters. As in Zarrinderakht et al. (2022), we assume that each crack propagates at a rate related to how much the stress intensity factor exceeds fracture toughness $K_{Ic}$ by

$$\dot{d}_{\mathrm{b}} = \max\left(-\frac{K_{\mathrm{Ib}} - K_{\mathrm{Ic}}}{K_{\mathrm{Ic}}|K'(0)|}, 0\right), \qquad \dot{d}_{\mathrm{t}} = \max\left(-\frac{K_{\mathrm{It}} - K_{\mathrm{Ic}}}{K_{\mathrm{Ic}}|K'(0)|}, 0\right), \tag{4}$$

where the overdot indicates differentiation with respect to time, and $|K'(0)|$ is the derivative of Freund's (1990) universal function $K$ (given by equation (6.4.26) in Freund's book), evaluated at zero crack propagation velocity. An approximate form of the universal function is $K(\dot{d}) \approx (1 - \dot{d}/v_{\mathrm{R}})/\sqrt{1 - \dot{d}/v_{\mathrm{p}}}$, with $v_{\mathrm{R}}$ and $v_{\mathrm{p}}$ being Rayleigh and primary wave velocities, so $-1/K'(0) \approx 2v_{\mathrm{p}}v_{\mathrm{R}}/(2v_{\mathrm{p}} - v_{\mathrm{R}})$. As discussed in Zarrinderakht et al. (2022), there are alternative hydrofracture-based models for crack tip propagation that could replace this description. We pursue the latter here due to the qualitative insights it provides.

For a given position of the cracks along the domain, the domain geometry is fully specified by ice thickness $H$, domain width $W$, and the crack lengths $d_{\mathrm{t}}$ and $d_{\mathrm{b}}$. In other words, we can treat the right-hand sides in equations (4) as being functions of $d_{\mathrm{t}}$ and $d_{\mathrm{b}}$, and we obtain a set of two coupled first-order differential equations. All that the complicated elastostatic problem described in Zarrinderakht et al. (2022) really does is to provide a means of computing $K_{\mathrm{Ib}}$ and $K_{\mathrm{It}}$ as functions of the dynamic variables $d_{\mathrm{t}}$ and $d_{\mathrm{b}}$.

In that vein, we will treat equations (4) as a two-dimensional dynamical system for $(d_\text{t}, d_\text{b})$. As in Zarrinderakht et al. (2022), our interest will be in steady state crack configurations, in identifying which sets of initial conditions lead to which final crack configurations, and in the effect of changing forcing and geometrical parameters on steady states and their basins of attraction. In particular, we want to identify what changes in parameter values lead to the disappearance of steady states in which the ice slab is only partially fractured so that cracks are either forced to propagate all the way across the ice or meet inside the ice. In either case, we will interpret the result as calving: the detachment of one side of the domain from the other.

To simplify the set of geometrical and forcing parameters, we non-dimensionalize the model using the same set of scales as in Zarrinderakht et al. (2022) and Lai et al. (2020). This leaves only the following dimensionless parameters,

$$\tau = \frac{R_{xx}}{\rho_\text{i} g H}, \qquad \eta = \frac{h_w}{H}, \qquad \kappa = \frac{K_{Ic}}{\rho_\text{i} g H^{3/2}}, \qquad W^* = \frac{W}{H}, \tag{5}$$

in addition to the dimensionless material constants given by Poisson's ration $\nu$, and

$$r = \frac{\rho_\text{i}}{\rho_\text{w}}. \tag{6}$$

Above, $\tau$ is a dimensionless extensional stress, $\eta$ a dimensionless depth to the surface water table, and $\kappa$ a dimensionless fracture toughness. We will primarily focus on dimensionless extensional stress $\tau$ and water level $\eta$ as forcing parameters, since $\kappa$ is likely small: with a dimesional fracture toughness $K_{Ic} = 0, 4$ MPa m$^{-1/2}$ (Rist et al., 1996) and an ice thickness of $H = 500$ m, $\kappa \approx 0.004$. To understand better how to map the dimensionless parameters to dimensional ones, recall that the extensional stress in an unconfined, one-dimensional ice shelf is $\rho_\text{i}(1-r)gH/2$ (van der Veen, 1983; MacAyeal and Barcilon, 1988). With a density ratio of $r = 0.89$, this corresponds to $\tau = 0.055$, which provides a reference value for the dimensionless extensional stress. The water level parameter is somewhat simpler: $\eta = 0$ corresponds to completely full surface cracks with the water level at the upper surface. $\eta = 1$ corresponds to a surface crack that remains dry no matter how far it is incised. A value of $\eta = 1 - r = 0.11$ represents a surface crack for which any portion below sea level is filled with water.

Using the Green's function formulation in Crouch and Starfield (1983), it can be shown that the solution for stress in the model in Zarrinderakht et al. (2022) is independent of Poisson's ratio $\nu$ (while displacements do depend on $\nu$). Consequently, the dimensionless stress intensity factors $K_\text{Ib}^*$ and $K_\text{It}^*$ depend only on the scaled crack lengths and on $(\tau, \eta, W^*, r)$. Equation (4) can therefore be written in the dimensionless form

$$\dot{d}_\text{b}^* = \max\left(K_\text{Ib}^*\left(d_\text{b}^*, d_\text{t}^*; \tau, \eta, W^*, r\right) - \kappa, 0\right), \qquad \dot{d}_\text{t}^* = \max\left(K_\text{It}^*\left(d_\text{b}^*, d_\text{t}^*; \tau, \eta, W^*, r\right) - \kappa, 0\right), \tag{7}$$

where the asterisk decorations on the crack length variables and stress intensity factors denote their dimensionless counterparts, scaled with ice thickness $H$ and with with the stress intensity $\rho_\text{i} g H^{3/2}$, respectively.

Note that, for simplicity, we immediately omit the asterisk decorations, in the understanding that all variables and parameters used below are dimensionless. Any changes in forcing parameters are assumed to occur much more slowly than cracks propagate, so the dimensionless forcing and geometry parameters $\tau$, $\eta$, $\kappa$, and $W$ are constant during crack propagation. The problem at hand is therefore an autonomous two-dimensional dynamical system, and we rely primarily on phase planes to visualize the behavior of the system: in the $(d_\text{b}, d_\text{t})$-plane, we plot curves traced out by $d_\text{t}(t)$ against $d_\text{b}(t)$ as time $t$ increases

for a solution to the dynamical system; these are the *orbits* of the dynamical system (7) (e.g. Strogatz, 1994). In practice, we use the boundary element method described in Zarrinderakht et al. (2022) to solve for the right-hand sides of equations (7) and use the MATLAB routine *streamslice* to generate phase planes.

Plotting orbits on a phase plane provides a simple graphical way of identifying the behavior of the system for a given set of parameters, for all possible initial conditions and a given set of parameter values. In that way, a phase plane is analogous to for instance Figure 10 of van der Veen (1998a), or Figures 4 and 7 in Zarrinderakht et al. (2022) for the single-crack systems considered in these papers. There, the evolution of the single crack length variable $d$ is determined graphically by plotting $K_{\mathrm{I}}(d)$: from this, one can read off whether a crack lengthens or not depending on whether or not stress intensity factor exceeds fracture toughness ($K_{\mathrm{I}} > \kappa$) or not. A perhaps even more direct analogue to a phase plane is shown in Figures 6 and 8b of Zarrinderakht et al. (2022), where (for a given set of parameter values), the range $0 \le d \le 1$ is divided into intervals for which $\dot{d} = 0$ and $\dot{d} > 0$, and hence indicates what state $d$ evolves. A phase plane generalizes this by not only indicating where $\dot{d}_{\mathrm{b}}$ and $\dot{d}_{\mathrm{t}}$ are positive and zero, respectively, but by showing the relative size of the rates of change, which determines the angle of the orbit and ultimately the state that the cracks evolve towards.

The ability to visualize evolution from arbitrary initial conditions using a phase plane also allows us to address how the dynamical system evolves under slow changes in forcing parameters (see also Zarrinderakht et al., 2022, sections 4-4–4.5): if started with a combination of forcing parameters that does not cause calving (generally with $\tau$ being too small or $\eta$ too large), partially incised crevasse will still typically result. A subsequent change in parameters may then lead to full crack penetration starting with initial conditions dictated by the previous formation of a partially incised crack (as opposed to short seed cracks only), subject to the caveat that we do not re-compute the full visocus pre-stress in this paper when doing so (but see also Zarrinderakht et al., 2023).

## 3 Results

### 3.1 Phase planes for aligned cracks

First, we generalize the geometry considered in Zarrinderakht et al. (2022) by introducing aligned surface and basal cracks at the mid-way point $x = W/2$ as in Figure 1. Note that the intention here is to mimic the infinitely wide domain of van der Veen (1998a,b) and Lai et al. (2020), and we use a large domain width $W = 10$, applying the same boundary conditions of vanishing elastic traction as Zarrinderakht et al. (2022).

Figure 2 shows an example of a $(d_{\mathrm{b}}, d_{\mathrm{t}})$-phase plane for this geometry. There are several non-standard qualitative features in this phase plane that turn out to be generic for the dynamical system (7), in each case related to the maximum over $K_{\mathrm{I}} - \kappa$ and zero taken on the right-hand side of equations (7). First, by construction, crack lengths can never shrink, so all orbits are either horizontal, vertical or angled upwards to the right: if a stress intensity factor is less than the fracture toughness, the corresponding crack tip simply does not move.

Second, the dynamical system is non-smooth: the maximum function on the right-hand sides of equations (7) not only ensures that cracks cannot shrink, it generally renders those right-hand sides non-differentiable where $K_{\mathrm{Ib}}^* = \kappa$ or $K_{\mathrm{Ib}}^* = \kappa$,

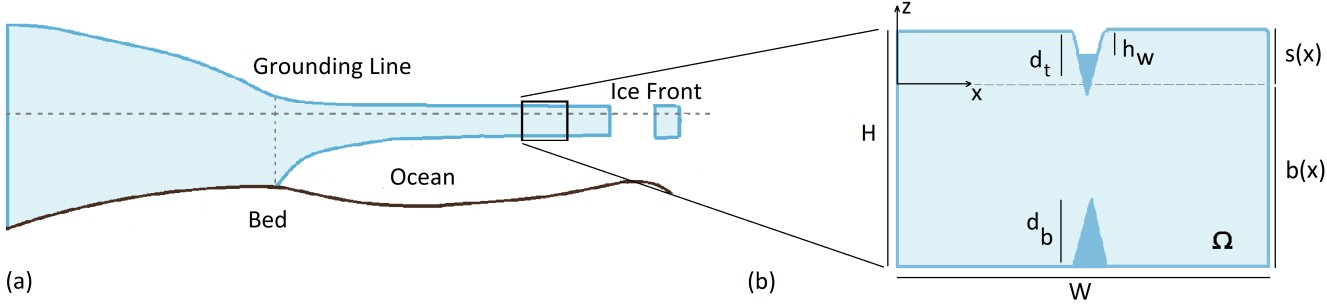

**Figure 1.** Geometry of the problem: the finite domain shown in panel (b) is intended to represent part of a floating ice shelf, with two aligned crevasses, one each at the upper and lower surfaces.

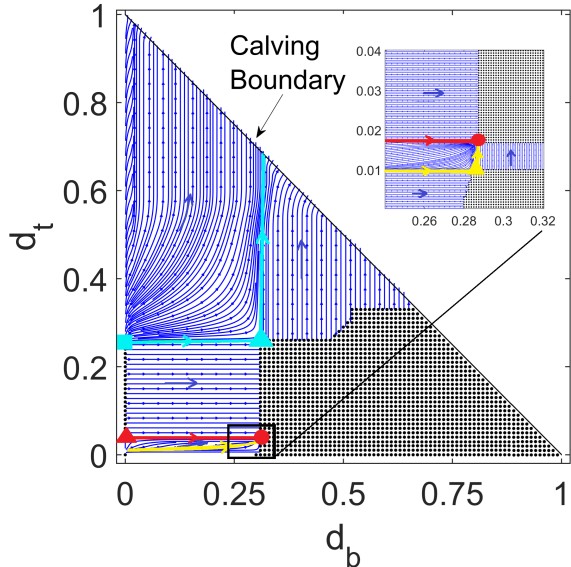

**Figure 2.** Phase plane diagram for two aligned cracks for $\tau = 0.02$, $\eta = 0.04$, $\kappa = 0.001$, $W = 10$ and $r = 0.89$. The inset shows an enlargement of the region marked as a black box around $d_b = 0.28$. Blue curves are orbits of the dynamical system (7), the direction in which $(d_b, d_t)$ evolves in time is indicated by blue arrowheads. The region of black dots indicates steady-state solutions for which both stress intensity factors are less than fracture toughness $\kappa$. The coloured dots indicate saddle-type (cyan, yellow and red triangles) and node-type (red circle and cyan square) marginal fixed points as defined in the main text. The cyan curves are the orbits into and out of the cyan saddle point, while the red and yellow curves are the orbits that delimit the basin of attraction of the red node point. A small region of steady states near the origin is not visible due to its small size, as is a thin strip of non-steady states close to the $d_b$-axis extending to the right of $d_b = 0.28$, discernible only in the inset.

even if $K_{\mathrm{Ib}}$ and $K_{\mathrm{It}}$ are smooth functions of $(d_{\mathrm{b}}, d_{\mathrm{t}})$ (where the latter seems likely unless a new contact area is formed, or a

190 section of open crack fully disappears at that point, see Figure 4a of Zarrinderakht et al. (2022)). For simplicity, we will refer to the sets of points for which $K_{\mathrm{Ib}}^* = \kappa$ or $K_{\mathrm{Ib}}^* = \kappa$ as marginal $d_{\mathrm{b}}$- and $d_{\mathrm{t}}$-nullclines, respectively: these are the curves along which one of the elastostatic stress intensity factors is equal to the fracture toughness. This awkward terminology is necessary here because a nullcline is simply a set of points for which $\dot{d}_{\mathrm{b}} = 0$ or $\dot{d}_{\mathrm{t}} = 0$; for smooth dynamical systems, these sets are generally one-dimensional curves, which is not the case here as we describe below. In a phase plane like Figure 2, parts of the

195 marginal nullclines are then the boundary between regions in which the orbits are purely horizontal or vertical (so $K_{\mathrm{It}}^* < \kappa$ or $K_{\mathrm{Ib}}^* < \kappa$, respectively), and "curved" orbits, along with both crack lengths change simultaneously. The change from curved to straight vertical or horizontal orbits is a graphical manifestation of the non-differentiability of the dynamical system.

Third, equilibria of the dynamical system are generally not isolated but occupy regions of finite size, rendered with black dots in Figure 2. These regions are again bounded by parts of the marginal nullclines defined above (where stress intensity

factors are equal to the fracture toughness). Here, the marginal nullclines separate regions in which orbits are either vertical or horizontal from regions of steady states in which all orbits are fixed points. Inside the regions of steady states, both stress intensity factors are less than the fracture toughness.

In addition, the phase plane here is bounded: crack lengths must be positive, and for aligned cracks, their sum must be less than the ice thickness. When using dimensionless crack lengths scaled with $H$, the crack tips meet when

$$d_{\mathrm{b}} + d_{\mathrm{t}} = 1, \tag{8}$$

which we take to correspond to calving as marked in Figure 2.

The usual notions of phase plane analysis, like identifying isolated fixed points and their stability, do not apply without modification due to the non-differentiability of the dynamical system, and due to the fact that equilibria occupy extended regions of the phase plane. Equilibria inside these extended regions are stable in the sense of Lyapunov but not asymptotically

stable (Strogatz, 1994): if perturbed, the state variable $(d_{\mathrm{b}}, d_{\mathrm{t}})$ stays nearby because it does not evolve. For equilibria on the boundary of a region of steady states (that is, equilibria on one of the marginal nullclines), we can distinguish between unstable and stable. The boundary is unstable there are orbits that point away from it, which is the case for boundaries at the top or to the right of a region of steady states, and stable (again in the sense of Lyapunov) otherwise.

There are several equilibria that occupy a special role, namely those where two marginal nullclines intersect. We will refer

to these equilibria as marginal fixed points below. There are five such equilibria in Figure 2, although one is only clearly visible in the enlargement in the inset.

One, marked with a red dot, is analogous to a stable node in standard phase plane analysis, and we will therefore refer to it as a "stable node" in a slight abuse of terminology. If we start the system with small surface and basal cracks of length $d_{\mathrm{t}}(0)$ and $d_{\mathrm{b}}(0)$, they will evolve towards this equilibrium solution provided the ratio $d_{\mathrm{t}}(0)/d_{\mathrm{b}}(0)$ is large enough and $d_{\mathrm{t}}(0)$ as well

as $d_{\mathrm{b}}(0)$ are not too small. The last caveat arises because, for the small values of $\kappa$ relevant to typical ice shelves, there is a small region around the origin in the phase plane (not visible in Figure 2 due to its small size) in which neither crack will grow. When we state that orbits started near the origin will evolve towards the node, we have to add that they need to start outside that

small region. The existence of short steady-state crack lengths has been discussed previously (van der Veen, 1998a; Lai et al., 2020), and is associated with low-stress intensity factors, scaling as $d^{1/2}\tau$ for the surface crack, and $d^{1/2}(\tau - 1 + r^{-1})$ for the basal crack (see appendix C1 of Zarrinderakht et al., 2022): for small enough $d$, these are guaranteed to be less than fracture toughness $\kappa$. An analogous region of very short steady-state cracks can be seen for instance in Figures 4 and 7 in Zarrinderakht et al. (2022).

Even though the stable node is not an attractor in the strict sense (there are other equilibria arbitrarily close to the stable node), it does have a finite basin of attraction demarcated by the red and yellow orbits into the stable node. Note that he size of that basin of attraction is easy to overestimate visually due to the finite resolution used in computing the phase portrait. Close to the stable node is a marginal fixed point that is analogous to a saddle in standard phase plane analysis, marked with a yellow triangle in the inset. For this marginal fixed point, a single orbit ends at the saddle, while a second orbit connects saddle and node (both shown in yellow in the inset). Below the orbit leading up to the saddle, there are additional orbits starting with lower values of $d_\mathrm{t}(0)/d_\mathrm{b}(0)$ that terminate at the boundary of a region of steady states as shown in the inset.

The third marginal fixed point is marked with a cyan triangle in Figure 2 and again plays the role of a saddle in a standard phase plane. Here, one orbit (marked in cyan) emerges from the saddle point towards the calving boundary, while a second orbit (also marked in cyan) connects a fourth (cyan square) marginal fixed point that is almost on the $d_\mathrm{t}$-axis to the cyan triangle saddle point; this orbit is analogous to a separatrix in a standard phase plane, and divides initial conditions that lead to immediate calving from initial conditions that lead to stable, steady cracks of finite length that leave the ice slab intact.

The unstable node point marked by a cyan square is paired with a third saddle point marked as a red triangle that is also located almost on the $d_\mathrm{t}$ axis ("almost" because there is in fact an imperceptible region near the $d_\mathrm{t}$-axis in which $d_\mathrm{b}$ does not grow, as discussed above). For the dynamics of a single top crack, the red triangle and cyan square points were previously identified elsewhere (see e.g., Figures 2a and 6a,b of Zarrinderakht et al. (2022)) as stable and unstable equilibria. That is, if we set $d_\mathrm{b} = 0$, then evolution of short cracks towards the smaller (red triangle) of the two being physically explained by the effect of the imposed tensile stress $\tau$ in opening the short crack eventually being overcome by cryostatic pressure as the crack lengthens. In fact, in all of the phase planes shown in the paper, the dynamics along each of the coordinate axes reduces to the dynamics of a single crack as previously discussed in Lai et al. (2020) and Zarrinderakht et al. (2022). As Figure 2 shows, the simple rationale regarding the stability of single cracks developed previously in these papers however falters when coupling the surface crack with a basal crack: for instance, both the red triangle and cyan square equilibria are actually unstable to the growth of a basal crack.

## 3.2 Changes in crack configuration due to altered forcing

Suppose the system is started with only small seed cracks to initiate crevasse growth (where these seed cracks need to be large enough in order to start outside the region of steady states around the origin discussed above). Calving will then occur if there is an orbit connecting the near-origin initial conditions in the phase plane to the calving boundary at $d_\mathrm{b} + d_\mathrm{t} = 1$. In Figure 2, that is not the case: instead, cracks will evolve to the configuration represented by the red dot, or one very close to it.

As in Zarrinderakht et al. (2022), we ask how the marginal fixed point at the red dot evolves under changes in forcing parameters, and how such changes can themselves lead to calving. The same caveat applies here as in our earlier work (see section 4.5 of Zarrinderakht et al. (2022)): in asking how the dynamical system changes under changes in parameters, and how an existing crack configuration is affected, we are ignoring the fact that the viscous pre-stress will generally not remain the of the same form as assumed in our model even over relatively short time scales (comparable with the Maxwell time of ice), and changes in domain geometry will occur as the result of viscous flow over long time scales so that results based on the rectangular geometry assumed here will eventually become misleading. We address these issues in the companion paper Zarrinderakht et al. (2023).

In Figure 3, we plot phase planes analogous to Figure 2 for a range of values of $\eta$ and $\tau$ as indicated for each column and row of the grid. Figure 2 is reproduced in panel c2. If we track the position of the (red) stable node-type marginal fixed point in Figure 2, we find that it moves upwards and to the right (that is, in the direction of flow of the dynamical system) under increases in extensional stress $\tau$, while it remains unchanged under an increase in $\eta$ (that is, an increase in the depth of the water table). In other words, if $\tau$ is increased, $(d_{\mathrm{b}}, d_{\mathrm{t}})$ will track the position of the node, but conversely, remain stranded in the region of steady states indicated by black dots under a subsequent reduction in $\tau$. This is analogous to becoming stranded in the grey region of steady states in Figure 6 of Zarrinderakht et al. (2022). The insensitivity to $\eta$ by contrast is easy to understand in terms of the shallow depth of the top crack $d_{\mathrm{t}}$ at the node: the tip of the surface crack is above the water level in columns 2–4 of Figure 3.

Calving under increases in $\tau$ eventually occurs for each of columns 2–4 through the node meeting the diagonal calving boundary in row a, with the surface crack length $d_{\mathrm{t}}$ remaining very shallow: calving occurs almost entirely by the propagation of the basal crack. The critical value of $\tau$ in each case lies somewhere between $\tau = 0.03$ and $\tau = 0.04$, which is consistent with the critical value of $\tau = 0.039$ for calving by basal crevasse propagation determined numerically in Zarrinderakht et al. (2022).

The saddle-type marginal fixed point marked by a cyan triangle moves downwards and to the right under increases in $\tau$, and moves downwards under decreases in $\eta$. An alternative calving mechanism, starting with a stable steady state configuration like the node in panel c2, is to raise the surface water level (that is, to decrease $\eta$) until the two marginal fixed points annihilate each other: this is analogous to the saddle-node bifurcations of Figures 6a and 6b of Zarrinderakht et al. (2022) (for instance, by going from panel c2 to panel c1 in Figure 3 here). When that annihilation occurs due to a reduction in $\eta$, the length of the basal crack remains almost unchanged during the subsequent evolution of $(d_{\mathrm{b}}, d_{\mathrm{t}})$: the orbit emerging from the saddle towards the calving boundary (shown in cyan in Figure 2 is nearly vertical, and hence the motion in the phase plane from the equivalent of a saddle-node bifurcation is also nearly vertical. Calving driven by changes in water level occurs almost entirely by the propagation of the surface crack.

In fact, if we suppose that a step from one panel in Figure 3 to a neighboring panel corresponds to calving (for instance, starting from the node in c2 and changing parameter values to those in c1, or starting from the node in panel b2 and changing parameter values to those in b1), then we can identify which crack will propagate to cause calving purely by looking at orbits that emerge near the origin in the panel in which calving occurs: in panel c1, orbits emerging near the origin evolve predominantly upward. More significantly still, if we start an orbit near the origin with $d_{\mathrm{b}} = 0$, that orbit will evolve to the

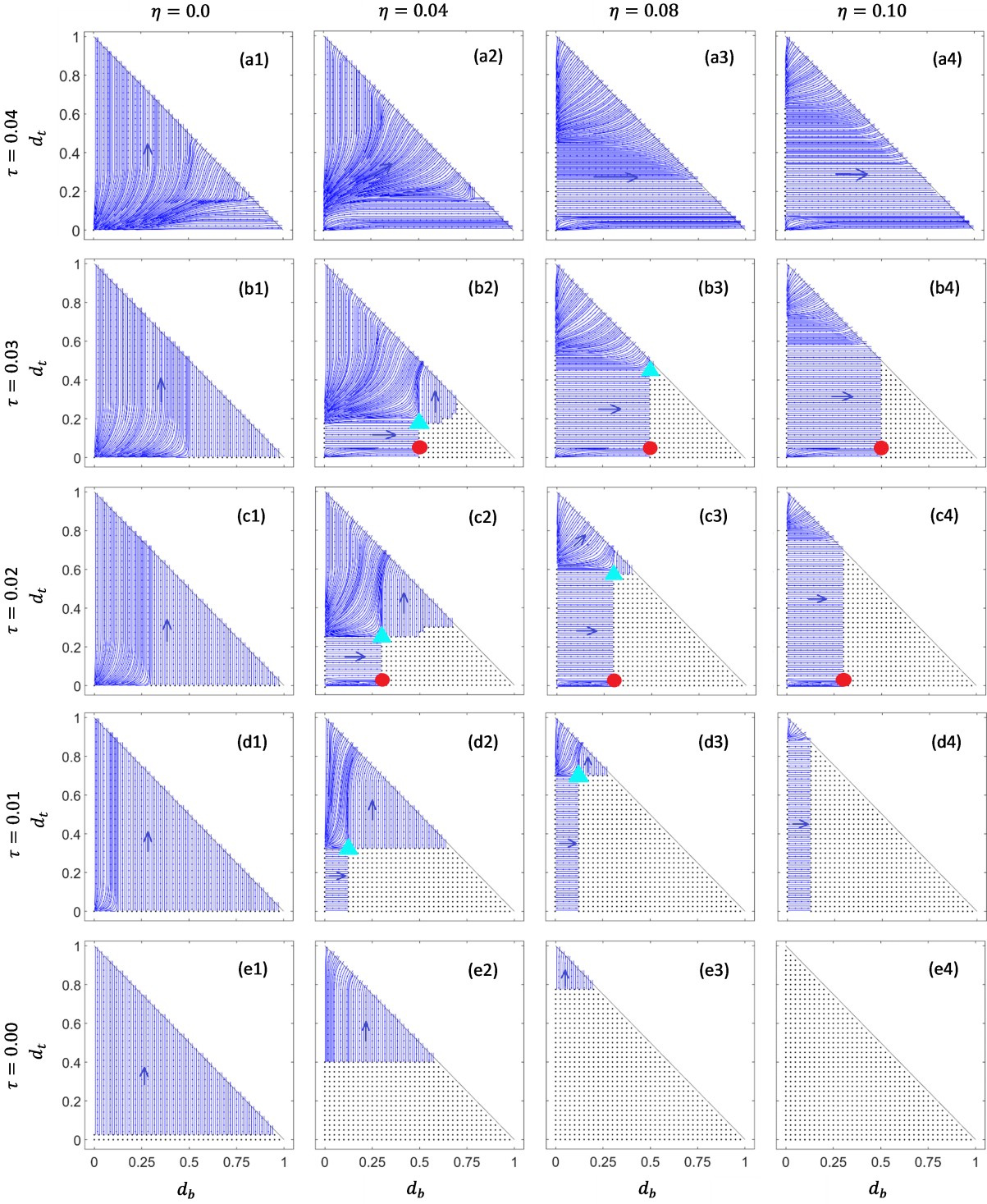

**Figure 3.** Gridded phase plane diagrams for two aligned cracks. $\eta = (0$ (column 1), 0.04 (column 2), 0.08 (column 3), 0.1 (column 4) and $\tau = 0.04$ (row a), 0.03 (row b), 0.02 (row c), 0.01 (row d), 0 (row e). The remaining parameters are those used in Figure 2.

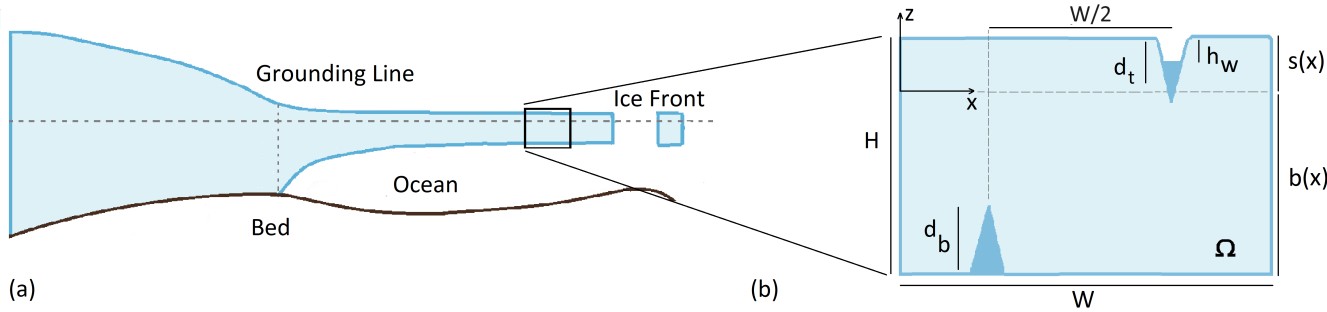

**Figure 4.** A periodic domain with offset cracks at $x = W/4$ and $x = 3W/4$.

calving boundary at $d_t = 1$ while maintaining zero basal crack length. Similarly, orbits emerging near the origin in panel b1 evolve predominantly to the right, and an orbit started near the origin with $d_t = 0$ will evolve to calving at $d_b = 1$ with zero surface crack length.

In other words, in order to predict parameter combinations that lead to calving with aligned cracks, we can actually look at the dynamics of surface and basal cracks in isolation: our admittedly coarse sampling of parameter space strongly suggests that calving occurs from near the origin (a nearly unfractured initial domain) as from the node configuration (with a short surface and larger basal crack). One crack is always dominant, and the propagation of one crack does not significantly reinforce the propagation of the other.

A plausible physical explanation of this behavior is provided by the torques generated on each crack face (see also Zarrinder-akht et al., 2022, appendix C2): the torque generated by extensional stress on and fluid pressure in the bottom crack induces a rotation on both sides of the crack (see also Figure 10 of Zarrinderakht et al. (2022)) that tends to compress the ice surface, and vice versa for the torque generated on a surface crack.

### 3.3 Offset crevasses in a periodic domain

Motivated by our conjecture that torques generated by a crack on one side of the domain affect the propagation of a crack incised into the opposite side of the ice, we consider whether the dynamics of misaligned (or laterally offset) cracks differ qualitatively from those of aligned cracks: presumably, if a basal crack causes compression at the upper surface, that compression is strongest above the basal crack and decreases with horizontal distance.

To maintain the requisite symmetry for laterally offset cracks to act as mode I crack and propagate vertically, we assume that they are located at $x = W/4$ and $x = 3W/4$, and switch to laterally periodic boundary conditions as in equations (3): as a result, the model describes a periodic array of surface and bottom cracks, in which the surface and bottom cracks are spaced a half-wavelength $W/2$ apart (Figure 4). Note that the assumption of vertical crack propagation is then consistent with a maximum hoop stress criterion (Zehnder, 2012, section 4.4.1), but we do not address the question of crack path stability (Cotterell and Rice, 1980), namely that a perturbed crack could evolve progressively away from a vertical orientation.

Figure 5 shows a grid of phase planes analogous to Figure 3. One obvious difference is that the "calving boundary" now consists of the lines $d_b = 1$ and $d_t = 1$: offset cracks will no longer meet part-way through the ice. We find similar behavior in Figure 5 as in Figure 3 for cases where crack evolution away from the origin does not lead to immediate calving, with a node-type marginal fixed point evident in panels d2–e2, c3–e3 and b4–e4. The corresponding saddle-type fixed point does not always exist, being absent in panels b4 and c4 (which is also the case in Figure 3, panels b4 and c4).

A subtle but significant qualitative difference with Figure 3 is the size of the surface crack at the node point: while basal crack length $d_b$ is significantly larger than surface crack length $d_t$, the latter is much bigger than in Figure 3. As a consequence, the tip of the surface crack corresponding to the node point equilibrium is below the water level and the stress field is actually affected by surface water pressure in several of the examples in Figure 5. As a result the node point moves upwards as $\eta$ is decreased (the crack lengthens as water level rises) in several examples (for instance, in going from panel 5c4 to panel 5c3, or panel 5d3 to panel 5d2).

This has a knock-on effect on the style of calving under increases in extensional stress: in the aligned crack case of Figure 3, an increase in extensional stress invariably leads to calving by basal crack propagation (see the discussion in section 3.2, and columns 2–4 of Figure 3). In the offset crack case of Figure 5, increases in $\tau$ by contrast lead to calving by top crack propagation through the annihilation of saddle and node points (panels d2–c2, c3–b3) more frequently than by the node point reaching the right-hand calving boundary (panels b4–a4), while the annihilation of saddle and node points in a saddle-node bifurcation only occurs as the result of a decrease in $\eta$ in Figure 3.

## 3.4 The effect of periodicity

The results in the previous section suggest that the presence of offset instead of aligned cracks may promote the simultaneous growth of both cracks: specifically, the motion of the node-type marginal fixed point under changes in parameter values involves significantly greater growth of the surface crack.

A second notable feature of Figure 5 as compared with Figure 3 is however that the values of extensional stress $\tau$ are significantly larger in the former than the latter, even though we obtain qualitatively similar phase planes (with a node and saddle type marginal fixed point in some of them, and therefore with parameter combinations that lead to finite crack growth but not to calving if starting near the origin). The fact that larger extensional stresses are required to produce a qualitatively similar result is not the result of having laterally offset versus aligned cracks, but of using a periodic domain with a relatively short ($W = 1$) length: in addition to surface and basal cracks interacting, neighboring cracks on each of the surfaces also interact.

Here we analyze the interaction between neighboring cracks, restricting ourselves to a periodic array of basal cracks for simplicity. The second equation in (7) then becomes redundant, and the dynamical system can be visualized simply by plotting $K_{Ib}$ against $d_b$, identifying where $K_{Ib} > \kappa$. We show results in Figure 6. Panel a shows how $K_{Ib}$ depends on crack length $d_b$ for fixed a value of $W = 2$ as $\tau$ is varied, analogously to Figure 7b in Zarrinderakht et al. (2022). For a given set of parameters, the crack will grow if $K_{Ib} > \kappa$ (the latter being the broken straight line), and remain static otherwise. Calving occurs if, as in the curves for $\tau = 0.12$ in Figure 6a, $K_{Ib} > \kappa$ up to the limiting value $d_b = 1$, at which the basal crack fully penetrates through

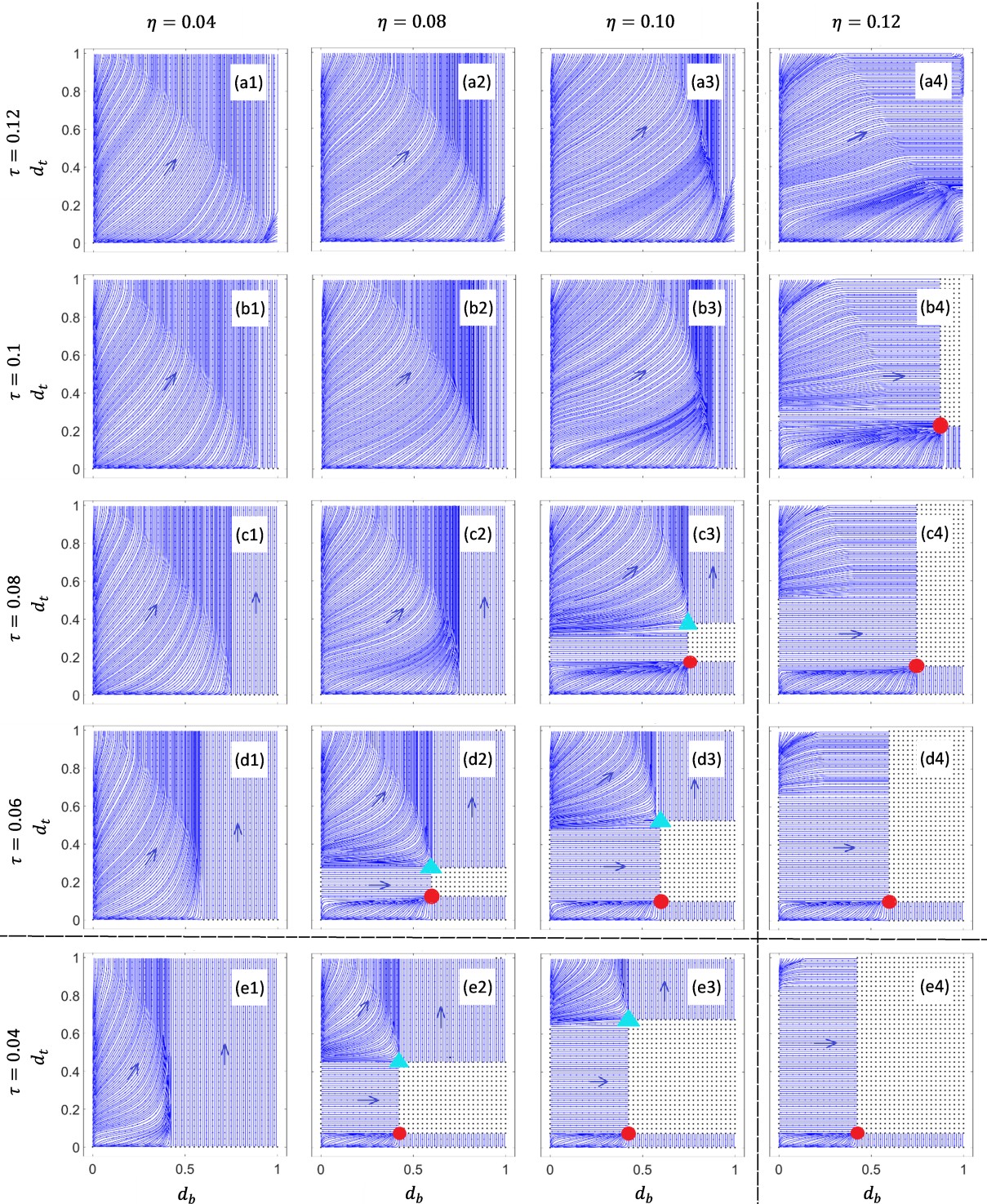

**Figure 5.** Gridded phase plane diagram for two offset cracks with periodic boundary conditions and $W = 1$, same plotting scheme as Figure 3. $\eta = 0.04$ (column 1), $0.08$ (column 2), $0.1$ (column 3), $0.12$ (column 4), and $\tau = 0.12$ (row a) $0,1$ (row b), $0.8$ (row c) $0.06$ (row d) and $0.04$ (row e). The vertical dashed line separates columns in which the surface water table is above sea level (to the left) from surface water tables below sea level (to the right). The horizontal dashed line separates rows in which $\tau$ *exceeds* the value of $0.055$ for an unconfined ice shelf (above) from those where $\tau$ is less than $0.055$ (below). All remaining parameter values are as in Figure 3.

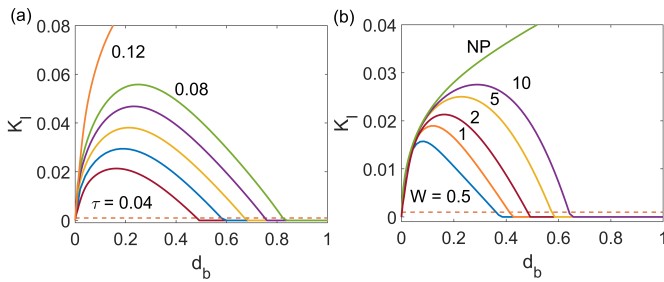

**Figure 6.** (a) Stress intensity factor $K_{\mathrm{Ib}}$ against crack length with $W = 2$ for a basal crack for different values of $\tau = 0.04$ (red), 0.05 (blue), 0.06 (yellow), 0.07 (purple), 0.08 (green) and 0.12 (orange). (b) Scaled stress intensity factor versus the crack length for different values of scaled periodic domain length, $W = 0.5$ (blue), 1 (orange), 2 (red), 5 (yellow), 10 (purple), and non-periodic domain shown by NP (green), for $\tau = 0.04$. The result for the non-periodic domain was computed for $W = 10$ but is insensitive to the domain width for traction-free lateral boundary conditions. The dashed horizontal line in both panels indicates the fixed parameter value $\kappa = 0.001$. The crack grows if $K_{\mathrm{Ib}} > \kappa$, and remains static otherwise.

the ice thickness. As shown in Figure 6b, this is the case for larger $\tau$, mimicking qualitatively the corresponding behavior for a non-periodic domain studied in Zarrinderakht et al. (2022).

Figure 6b similarly shows $K_{\mathrm{Ib}}$ against crack length for a fixed value of $\tau = 0.04$, just above the critical value of 0.039 for calving by basal crack propagation for a non-periodic domain in Zarrinderakht et al. (2022). The curve marked "NP" reproduces the corresponding result for that domain, given by the purple curve in Figure 7b of Zarrinderakht et al. (2022). If we compare the case of a finite-width periodic domain with the case of the non-periodic domain, we find that $K_{\mathrm{Ib}}$ agrees between the two for short crack lengths $d_{\mathrm{b}}$, for which the far field is relatively unimportant. The difference becomes noticeable as $d_{\mathrm{b}}$ grows,
more so for short periodicities $W$. Only the non-periodic case experiences calving at $\tau = 0.04$ in Figure 6b.

For each combination of parameters $\tau$ and $\kappa$ and a given domain periodicity $W$, the points of intersection between the corresponding solid curve and the dashed curve in Figure 6 correspond to marginal equilibria as defined in section 3.1, with the smaller solution being unstable and the larger being stable if it exists. As extensional stress is increased, the larger, stable marginal equilibrium also increases, until it disappears entirely. Figure 7a shows the corresponding steady states for $W = 10$
as a continuous function of the parameter $\tau$, by analogy with Figure 8b in Zarrinderakht et al. (2022). The grey region consists of steady states for $W = 10$ as an example to clarify that the curves shown separate regions of steady states from regions of phase space in which $d_{\mathrm{b}}$ increases.

As Figure 7a shows, calving for basal cracks occurs through the continuous growth of crevasse length with increasing $\tau$ until $d_{\mathrm{b}} = 1$ is reached at a critical value $\tau_{\mathrm{crit}}$. For the case $W = 10$ shown in Figure 6a, $\tau_{\mathrm{crit}} = 0.088$, as compared with
$\tau_{\mathrm{crit}} = 0.039$ for a non-periodic domain. In Figure 7b, we plot the dependence of $\tau_{\mathrm{crit}}$ on $W$: the larger $W$, the lower the critical stress. We conjecture (but have not attempted to prove) that the limit of $\tau_{\mathrm{crit}}$ for large $W$ is given by the critical stress for a non-periodic domain with traction-free lateral boundaries, $\tau_{\mathrm{crit},\infty} = 0.039$. If so, the decrease in $\tau_{\mathrm{crit}}$ clearly continues to be significant beyond the range for which we have calculated $\tau_{\mathrm{crit}}$ in Figure 7: for $W = 20$, $\tau_{\mathrm{crit}}$ is a little under twice of

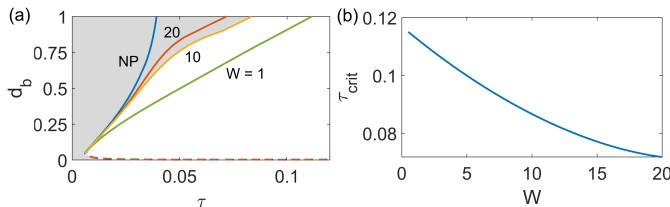

**Figure 7.** (a) Bifurcation diagram for a periodic domain with a single basal crack with $W = 1$ (green), 10 (yellow), 20 (red) and non-periodic domain shown by NP (blue). The region of steady states generally lies to the left of each curve, as indicated by grey shading for $W = 10$. (b) The critical extensional stress as a function of the domain width $W$.

the critical value for a non-periodic domain. Note that in the limit of a small crevasse spacing (much less than a single ice thickness), the effect of neighbouring crevasses observed here agrees with the previous results of Weertman (1973) and van der Veen (1998a), who found a significant reduction in crack tip stress intensity factor for crevasses that are spaced closer than their depth of penetration, relative to an isolated crevasse.

As in the non-periodic case considered in Zarrinderakht et al. (2022), calving occurs because $K_{\mathrm{Ib}}$ changes from vanishing as $d_{\mathrm{b}} \to 1$ for $\tau < \tau_{\mathrm{crit}}$ to becoming infinite for $\tau > \tau_{\mathrm{crit}}$ (Figure 6a). In Zarrinderakht et al. (2022, appendix C), we were able to associate that behavior with the net torque generated by the crack on the remaining "neck" of ice immediately above the crack. In a non-periodic domain with vanishing lateral traction, the torque generated by forces on the crack face must be balanced by a torque generated by stresses in the neck of ice. The balance of torques dominates the stress field in the neck when the crack spans nearly the full ice thickness, and a positive stress intensity factor (diverging to $+\infty$ as $d_{\mathrm{b}} \to 1$) results if the net torque on the crack remains positive for basal crack lengths approaching unity.

In the present case, the torque generated by one crack is balanced not only by the torque in the overlying neck of ice but also by torques due to all the other cracks in the periodic array, which reduces the net torque and therefore stresses generated in the neck of ice. The smaller the spacing $W$ between cracks, the stronger that effect becomes, which we can visualize by looking at the vertical displacement $u_3$ around the crack. Figure 8 shows the stress field $\sigma_{xx}$ and corresponding deformation of the domain for identical parameter values $\tau = 0.1$.

Clearly, stresses are larger around the crack tip for the non-periodic case in panel a. Far-field displacements $u_z$ displacement are also larger for the non-periodic case, and linear in $x$, without any applied traction at the lateral boundaries, the far field undergoes a rigid body motion. By contrast, the periodic case has smaller far-field displacement with a finite, negative curvature: far from the crack, the stress and displacement fields are those for an elastic beam, with the curvature signaling a bending moment that resists the torque generated in the crack.

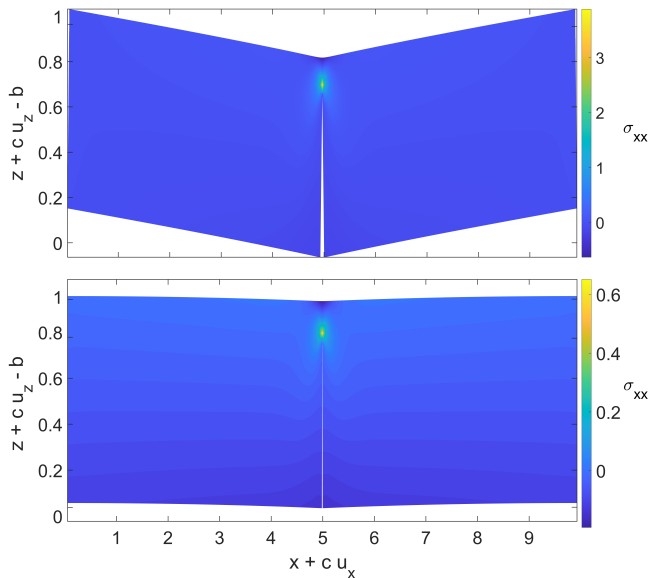

**Figure 8.** The stress distribution for a basal crevasse with $\tau = 0.1$ and $d_{\mathrm{b}} = 0.8$ for (a) a non-periodic domain with $W = 10$, (b) a periodic domain with $W = 10$. As in Figure 10 of Zarrinderakht et al. (2022), we illustrate the results here using a deformed domain, plotting $\sigma_{xx}$ against Eulerian position $(x + cu_x, z + cu_z)$ with $c = 0.02$.

## 4 Discussion

### 4.1 Towards a calving law

As in the simpler case of a single crack studied by van der Veen (1998a,b), Lai et al. (2020) and Zarrinderakht et al. (2022), we can attempt to create a calving law for the case of interacting cracks incised into opposite sides of the ice. Identifying which parameter combinations $(\tau, \eta, \kappa)$ result in calving given an initial state with only minimal crack lengths$(d_{\mathrm{b}}(0), d_{\mathrm{t}}(0))$ is relatively simple. First, the initial crack lengths need to be sufficient for the stress intensity factor to overcome the typically small fracture toughness $\kappa$. As the crack lengths are short, the effect of interactions between cracks is minimal in that case: the results of van der Veen (1998a,b), Lai et al. (2020) and Zarrinderakht et al. (2022) suffice, with both Lai et al. (2020) and Zarrinderakht et al. (2022) furnishing the relevant criteria. Second, the ensuing crack propagation needs to result in the cracks either meeting inside the ice as in section 3.2, or one crack propagating fully through the ice as in section 3.3. That outcome is easy to identify through the absence of the node-type marginal fixed point in the relevant phase planes, as shown in Figure 3 and 5. The relevant region of parameter space can be delineated crudely from Figures 3 and 5 directly, which could be improved upon by identifying conditions under which the fixed point marginally exists (defining the boundary of the region of parameter space in which calving occurs): for instance, when it first appears at the "calving boundary" (the diagonal $d_{\mathrm{b}} + d_{\mathrm{t}} = 1$ in Figure 3, or the right-hand edge $d_{\mathrm{b}} = 1$ in Figure 5), or when it first forms at a saddle-node bifurcation (for instance, between the parameter values used in panels d2 and c2 of Figure 5).

A more complicated problem is the construction of a calving law that allows for slow changes in the forcing parameters: by slow, we mean as compared with the natural time scale of crack evolution in the dynamical system (7), which is likely the case in any realistic setting, where extensional stress $\tau$ and water level $\eta$ typically evolve on the advective time scale for ice transport across an ice shelf, and on the seasonal time scale associated with surface melt, respectively. In that situation, cracks

may propagate only part-way across the ice. Calving can occur because of subsequent changes in forcing parameters to cause them to propagate further and eventually meet in the ice or reach the opposite ice surface.

In the confines of our model, the difference between calving by such slow changes in forcing parameters, and instant calving as discussed above is that the cracks will have a significant initial length at the instant that a critical parameter combination permitting calving is reached. The analogous situation for a single crack was discussed in section 5 of Zarrinderakht et al.

(2022) and led to a relatively simple prescription for calving (equation (37) of Zarrinderakht et al. (2022)), but the present case is more complex to capture.

Consider the case of offset cracks as shown in Figure 5. The simplest approximation is to assume that the region of steady states indicated by black dots as in panels d2–e2, c3–e3 and b4–e4 always takes the form of a rectangle with its sides aligned with the $(d_\mathrm{b}, d_\mathrm{t})$-axes. Let the upper and lower boundaries of the region be at $d_\mathrm{tm}^+(\tau, \eta, \kappa)$ and $d_\mathrm{tm}^-(\tau, \eta, \kappa)$, and the left-hand

boundary at $d_\mathrm{bm}(\tau, \eta, \kappa)$. If any one of these boundaries is not present (as is the case for $d_\mathrm{tm}^+$ in panels b4 and c4), then simply set the corresponding value of $d_\mathrm{.m}$ to unity.

The marginal node-type marginal fixed point, if present, must be at $(d_\mathrm{tm}^-, d_\mathrm{bm})$, and any initial evolution is to that fixed point, at least to a close approximation. Subsequent movement of the boundaries of the rectangle of steady states can have one of four consequences. First, the node can move upwards and to the right, and in which case the crack configuration $(d_\mathrm{b}, d_\mathrm{t})$ tracks the

425 location of the node. Second, the node can move downwards and to the left, leaving the crack configuration $(d_\mathrm{b}, d_\mathrm{t})$ static and inside the region of steady states. Third, the node can move down and to the right, in which case $d_\mathrm{b}$ tracks the left-hand edge $d_\mathrm{bm}$ of the region of steady states, while $d_\mathrm{t}$ remains static. Forth, the node can move upwards and to the left, in which case $d_\mathrm{t}$ tracks the bottom of the region of the steady states at $d_\mathrm{tm}^-$ while $d_\mathrm{b}$ remains static.

Any movement of the boundaries of the region of steady states at $d_\mathrm{tm}^+$, $d_\mathrm{tm}^-$, and $d_\mathrm{bm}$ that occurs later leads to a similar

pattern of crack evolution: $(d_\mathrm{b}, d_\mathrm{t})$ will either move horizontally to stay on the left-hand edge of the region of steady states, move vertically to stay on the bottom edge, move diagonally to track the corner that is the node-type marginal fixed point or remain inside the region of steady states if none of the other alternatives are possible. That is until either $(d_\mathrm{b}, d_\mathrm{t})$ reaches the right-hand edge of the unit square and calving occurs by the propagation of the bottom crack all the way to the upper ice surface, or the upper boundary of the region of steady states reaches the location $(d_\mathrm{b}, d_\mathrm{t})$. The latter outcome can occur either

as a result of the upper boundary reaching a location $(d_\mathrm{b}, d_\mathrm{t})$ that had previously ended up on the inside of the region of steady states or because the upper boundary $d_\mathrm{tm}^+$ reaches the lower boundary at $d_\mathrm{tm}^-$ and the two marginal fixed points annihilate each other in a saddle-node bifurcation. In either event, calving occurs by the propagation of the surface crack to the base of the ice.

Mathematically, we can capture the implied motion of the phase point $(d_\mathrm{b}, d_\mathrm{t})$ under changes in parameter values by the following statement, analogous to equation (37) in Zarrinderakht et al. (2022): let

$$S_\mathrm{b}(\tau, \eta, \kappa) = \{d : d > d_\mathrm{bm}(\tau, \eta, \kappa)\}, \qquad S_\mathrm{t}(\tau, \eta\kappa) = \{d : d_\mathrm{tm}^-(\tau, \eta, \kappa) \leq d < d_\mathrm{tm}^+\} \cup \{1\}. \tag{9}$$

Then, if $T$ is a slow time variable associated with the change in parameters (as opposed to the fast time variable with respect to which the dynamical system (7) evolves), then we can write

$$d_{\rm b}(T) = \inf\{d : d \in S_{\rm b}(\tau(T), \eta(T), \kappa(T) \text{ and } d > d_{\rm b}(T') \text{ for all } T' < T\}, \tag{10}$$

$$d_{\rm t}(T) = \inf\{d : d \in S_{\rm t}(\tau(T), \eta(T), \kappa(T) \text{ and } d > d_{\rm t}(T') \text{ for all } T' < T\}; \tag{11}$$

all that is then required is to know how the boundaries of the region of steady states depend on $\tau$, $\eta$, and $\kappa$.

## 4.2  Limitations of the model

We have not attempted to complete the calculation suggested at the end of the last subsection, namely to compute in detail how the boundary of the region of steady states depends on forcing parameters. The reason is that Figure 5 (on which the procedure described was based) was computed for a specific value $W$ of the spacing between crevasses, and we demonstrated
in section 3.4 that our results are highly sensitive to that spacing. More specifically, we found no preferred finite spacing between crevasses at which calving occurs most easily (Figure 7b). It appears that basal crevasses propagate more easily the further apart they are spaced, as the torques exerted on each other by neighboring cracks have the effect of acting to close the tracks (Figure 8), and this makes the exercise of trying to define a calving rule for finitely-spaced crevasses an exercise of limited value.
The absence of an optimal, finite crevasse spacing at which crack propagation occurs most easily (in the sense of calving occurring at the lowest possible extensional stress $\tau$) likely points to the same deficiency in the model as previously discussed in section 6.3 of Zarrinderakht et al. (2022): based on the assumption of small-strain elasticity, the model neglects the effect of vertical elastic displacements on fluid pressure at the domain boundary. In other words, the effect of elastic deformation on buoyancy is excluded from the model, even though at large lateral distances, we can expect sufficient vertical displacements to have a leading order effect. We anticipate that incorporating the feedback between displacement and fluid pressure at the boundary will lead to additional torques generated by vertical displacements in the far field, suppressing crack growth for very large crack spacings (see also Buck and Lai, 2021). We leave a study of this effect to future work.

There are several other limitations in addition to not accounting for the effect of buoyancy on elastic stresses. For a given elastic pre-stress, the linear elastic fracture mechanics problem solved here relies on the same weakly inertial propagation rate
prescription due to Freund (1990) that was previously used in Zarrinderakht et al. (2022). Since the cracks under consideration are typically fluid-filled, it is likely that dynamic propagation is controlled by the retarding effect of fluid flow in the fractures Spence et al. (1985), which require a significantly more complicated hydrofracture model, which is unlikely to permit a comprehensive study of parameter space, or even of fracture evolution for different initial conditions as in Figures 2, 3 and 5. In addition, the assumption of purely vertical crack propagation is contingent on the highly specific crack orientations consid-
ered here, which ensure that we have purely mode 1 crack propagation. In reality, there are likely to be many interacting and potentially curved cracks, which we will address with a future iteration of the model.

There are two other major complications that need to be addressed. The first is the computation of the viscous pre-stress itself. In the present work, we have insisted on a parallel-sided slab as the basic domain into which cracks are incised, with

viscous pre-stress that corresponds to a completely unfractured ice slab. In practice, we expect these assumptions to break

down over different time scales: first, once cracks have propagated to a steady state configuration in which they span only part of the ice thickness, the elastic stress built up during fracture propagation will decay viscoelastically over a single Maxwell time, likely on the order of hours in an ice shelf (Olinger et al., 2022). After that viscoelastic relaxation, the form of the pre-stress assumed here as well as in van der Veen (1998a,b), Lai et al. (2020) and Zarrinderakht et al. (2022) will no longer be appropriate; a more complicated recomputation of the viscous pre-stress on the now partially fractured domain becomes

necessary (Krug et al., 2014). Second, over the much longer time scale associated with significant viscous deformation of the ice shelf (typically decades or longer), even the fractured ice domain will no longer take the form of a parallel-sided slab with narrow cracks incised into them: the cracks will be widened by viscous flow, significantly altering the domain on which the linear elastic fracture mechanics problem considered here needs to be solved. We consider both of these complications in the companion paper (Zarrinderakht et al., 2023).

The second major issue is the prescription of surface hydrology through a surface water table elevation, as we do through the parameter $\eta$. In reality, near-surface aquifers or surface hydrological systems are unlikely to be spatially uniform, or unaffected by the evolution of the crevasses that they feed: drainage of water into a crevasse should introduce an additional dynamic timescale and set of processes that the prescription of a fixed water level completely ignores. This is unlikely to be a trivial issue, as ice shelves are unlikely to be isothermal even if the near-surface supports an aquifer: if water is supplied slowly and

crevasse propagation is slowed, then refreezing of water in a newly formed crack (a process we ignore here on the basis that we assume rapid crack propagation) could suppress crevasse growth.

A simpler end member is, of course, the case of a dry ice shelf surface ($\eta = 1$ in our notation). This is not a case that we have dwelt on, as we have focused on the richer dynamics generated by a finite water table depth. The rightmost columns of Figures 3 and 5 however provide a guide to what happens as we keep lowering the water table: surface crack propagation

is progressively inhibited. For aligned cracks as in Figure 5, a small surface crack generally forms but its tip remains above water level unless initial conditions to the contrary are specified, and calving happens due to bottom crevasse propagation at sufficiently large extensional stress $\tau$. For offset cracks, the top crack is in column 4 of Figure 5 still penetrates below the water level, but the actual depth to which it penetrates appears to have no bearing on steady state bottom crack lengths, and calving once more occurs due bottom crevasse propagation at sufficiently large $\tau$. In other words, for progressively drier top cracks

(meaning, larger values of $\eta$ approaching unity), we expect a model for bottom crevasse formation only to predict calving adequately.

## 5   Conclusions

In this study, we modeled the simultaneous growth of both basal and surface crevasses which has received less attention in the literature due to a lack of tabulated Green's functions. Here a semi-smooth two-dimensional dynamical system is used

to display crack growth rate, $(\dot{d}_b, \dot{d}_t)$, and their interactions as a phase plane with the constraints imposed by the contact conditions. The orbits in this system show the direction of propagation flows with arrows. In other words, it shows the rate of

evolution of crack length as a function of the current crack length for a fixed set of parameters. The steady regions are shown by dots that no crack penetration occurs. This model can predict the evolution of cracks to either a steady state or to the point where the cracks span the width of the ice, known as calving, for a given set of initial conditions and parameters.

For two aligned crevasses in the shelf, we observed that the top crack dominates when the water level is high and there is a significant amount of water in it. When the shelf is intact, the basal crack grows rapidly and reaches a steady state where the hydrostatic pressure is equal to the cryostatic pressure. Meanwhile, the surface crack, despite containing a large amount of water, remains small and stable until a certain point and then becomes unstable and grows to span the entire thickness of the shelf. Our study has shown that no set of parameters can result in both cracks growing simultaneously from an undamaged

shelf to calving while considering only one crack is insufficient to cause calving. The model indicates that one crack is always dominant, and the scenario of two aligned cracks growing at the same time to a steady state is not predicted. This behavior can be attributed to the difference in crack growth speed and stress intensity factor at the crack tips. The results support the conclusion that the calving criteria established by Zarrinderakht et al. (2022) is effective for the case of two aligned crevasses and does not require any modification to the system behavior.

Driven by our hypothesis that torques produced by a crack on one side of the domain can impact a crack on the opposite side of the ice, we investigated laterally offset cracks. To ensure that misaligned cracks maintain the necessary symmetry to function as a mode I crack and propagate vertically, we assume that they are located at regular intervals. Considering periodic boundary conditions and the frequency of cracks on a shelf can improve stability and resistance to crack propagation under higher extensional stress. In this case, the domain length and crack spacing become effective, and as the domain grows and the

crack separation increases, the stress intensity factors also increase. The model proposed in this study, similar to Zarrinderakht et al. (2022), does not account for the buoyancy effect that generates a torque to stabilize the crack, resulting in a tilt in the domain. However, periodic boundary conditions counteract this bending effect by reproducing the crack pattern as the shelf bends back down to the next crack. In addition, applying periodic boundary conditions causes steady state and fixed points to occur at later stages of larger crack lengths.

*Code availability.*  The code for calculations is available from the corresponding author upon request.

*Author contributions.*  MZ executed the research. MZ and CS designed the project and wrote the paper. MZ and AP developed the numerical method and the code. All authors edited the paper.

*Competing interests.*  The authors declare no competing interests

*Acknowledgements.* MZ was supported by the ArcTrain NSERC CREATE graduate training program and by NSERC Discovery Grant funding to CS. CS acknowledges NSERC Discovery Grant RGPIN-2018-04665. AP acknowledges the support of NSERC Discovery Grant RGPIN-2015-06039.

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
