# Peer review of "An analysis of the interaction between surface and basal crevasses in ice shelves"

_EGUsphere, 2023_

## Author Comment (AC1)

**Response to RC1: 'Comment on egusphere-2023-1252', Anonymous Referee #1, 08 Aug 2023'**

January 30, 2024

**Reviewer Comment:** Determining the fate of surface and basal cracks on ice shelves is an important yet unsolved problem in the cryosphere. In this paper, following a prior work focused on the development of the boundary element model and LEFM by the same authors, a novel dynamical system approach is employed to generalize the predicted outcomes of surface and basal cracks for a range of relevant parameters. The final result is subject to a few assumptions, such as the periodic boundary condition, and the neglect of the buoyancy restoring force related to the ice-ocean boundary condition. However, the dynamical system analysis and interpretation of the phase space are generally applicable pending improvements to the LEFM model itself. My comments primarily pertain to the clarity of the model explanations. With these minor revisions, I recommend this paper for publication.

First, as this is the first time a phase plane approach for crack propagation is presented, the authors can include some examples where clear theoretical expectations exist to facilitate the explanation of the phase plane. For example, the authors mention that their previous analytical result that $\tau = 0.039$ would cause calving by basal crevasse. Theoretically without surface water we would simply see zero surface crack growth and basal crack growth. I think Fig 3 a4 attempted to illustrate this, but I'd think for $\eta = 0.1$, any surface crack depth $< 0.1$ corresponds to dry surface crack. Thus $d_t < 0.1$ would all correspond to arrows pointing towards the right. Could the authors explain the curved trajectories near $[dt, db] = [0, 0]$?

**Response:** If we understand the comment correctly, then it appears to say is that a dry surface crack should not be able to propagate. That would, however be incorrect: consulting figure 6a of Zarrinderakht et al (2022; cited in the present manuscript), we can see that at $\tau = 0.04$, there is a finite region of non-steady surface cracks lengths near $d_t = 0$ even in the absence of basal cracks, between the dashed and solid maroon lines in that figure. Physically, these cracks grow because for relatively short cracks the extensional stress $\tau$ acting on the crack faces is able to overcome the effect of cryostatic pressure and generate a positive stress intensity factor even when the crack is dry.[1]
* * *
[1] A minimal crack length is however required to elevate that intensity factor to or above the fracture toughness; that short length is too small to be rendered in the phase diagrams of the present paper, but corresponds to the region between the horizontal axis and the dashed maroon line of figure 6a of Zarrinderakht et al 2022. We refer to this small region in the present paper when we say in section 3.1,

*The last caveat arises because, for the small values of $\kappa$ relevant to typical ice shelves, there is a small region around the origin in the phase plane (not visible in Figure 2 due to its small size) in which neither crack will grow. When we state that orbits started near the origin will evolve towards the node, we have to add that they need to start outside that small region. The existence of short steady-state crack lengths has been discussed previously (van der veen 1998a, Lai et al 2020), and is associated with low-stress intensity factors, scaling as $d^{1/2}\tau$ for the surface crack, and $d^{1/2}(\tau - 1 + r^{-1})$ for the basal crack (see appendix C1 of Zarrinderakht et al, 2022): for small enough d, these*

Of course, make the dry surface crack too long, however, and cryostatic pressure will cause the stress intensity factor to go to zero. These observations would account for why orbits along the $d_t$-axis point up for fairly small $d_t \lesssim 0.1$ in figure 3a4 of the present submission, in line with figures 2a and 6a of Zarrinderakht et al (2022).

We have amended the text in two places to tie these concepts together. First, at the end of the revised section 2, we expand on phase place plotting by linking the information contained in phase plane plots with the simpler ways of graphically displaying how cracks grow in previous papers on single cracks:

*Plotting orbits on a phase plane provides a simple graphical way of identifying the behavior of the system for a given set of parameters, for all possible initial conditions and a given set of parameter values. In that way, a phase plane is analogous to for instance to Figure 10 of van der Veen(1998a), or Figures 4 and 7 in Zarrinderakht et al (2022) for the single-crack systems considered in these papers. There, the evolution of the single crack length variable d is determined graphically by plotting $K_I(d)$: from this, one can read off whether a crack lengthens or not depending on whether or not stress intensity factor exceeds fracture toughness ($K_I > \kappa$) or not. A perhaps even more direct analogue to a phase plane is shown in Figures 6 and 8b of Zarrinderakht et al (2022), where (for a given set of parameter values), the range $0 \le d \le 1$ is divided into intervals for which $\dot{d} = 0$ and $\dot{d} > 0$, and hence indicates what state d evolves. A phase plane generalizes this by not only indicating where $\dot{d}_b$ and $\dot{d}_t$ are positive and zero, respectively, but by showing the relative size of the rates of change, which determines the angle of the orbit and ultimately the state that the cracks evolve towards.*

To address the "clear theoretical expectations" part of the point raised by the reviewer (modulo the explanation of why the clear theoretical expectations may differ from those the reviewer has in mind), we have amended the end of section 3.1 to address specifically the behaviour on (or very near) the coordinate axes of the phase plane, pointing out that the dynamics of the system along the coordinate axes is exactly that predicted by previous studies of single cracks, but also making clear that predictions of stability and instability for those single cracks may fail when coupled with another crack:

*The third marginal fixed point is marked with a green dot in Figure 2 and again plays the role of a saddle in a standard phase plane. Here, one orbit (marked in green) emerges from the saddle point towards the calving boundary while a second orbit (also marked in green) connects a fourth (cyan) marginal fixed point that is almost on the $d_t$-axis to the green saddle point; this orbit is analogous to a separatrix in a standard phase plane, and divides initial conditions that lead to immediate calving from initial conditions that lead to stable, steady cracks of finite length that leave the ice slab intact. The cyan-coloured unstable node point is paired with a third saddle point marked in magenta that is also located almost on the $d_t$ axis ("almost" because there is in fact an imperceptible region near the $d_t$-axis in which $d_b$ does not grow, as discussed above). For the dynamics of a single top crack, the magenta and cyan points were previously identified elsewhere (see e.g., figures 2a and 6a,b of Zarrinderakht et al 2022) as stable and unstable equilibria, evolution of short cracks towards the smaller (magenta) of the two being physically explained by the effect of the imposed tensile stress $\tau$ in opening the short crack eventually being overcome by cryostatic pressure as the crack lengthens. In fact, in all of the phase planes shown in the paper, the dynamics along each of the coordinate axes reduces to the dynamics of a single as previously discussed in Lai et al (2020) and Zarrinderakht et al (2022). As figure 2 shows, the simple rationale regarding the stability of single cracks developed previously in these papers however falters when coupling the surface crack with a basal crack: for instance, both the magenta and cyan equilibria are in fact unstable to the growth of a basal crack.*
* * *
*are guaranteed to be less than fracture toughness $\kappa$.*

**Reiewer Comment:** Would you have a simpler phase plane if water height is rescaled by the crack depth rather than ice thickness?

**Response:** Assuming that scaled water height is to be treated as a constant, then scaling the physical water height by crack depth corresponds to different physics (a water table whose location changes as the crack evolves, in such a way that the water table is always at a certain fraction of the crack length. It's possible this would elad to a simpler phase plane, depending on the definition of "simpler". However, we struggle to envisage a physical situation in which the water table would evolve in this way. A perhaps easier to justify alternative to our fixed water table elevation would be a fixed water volume in the crack. As discussed in Zarrinderakht et al (2022), that situation however comes with a number of awkward caveats, and as a result, we have not considered it here.

**Reviwer Comment:** On the other hand, when the surface crack is fully filled with water, and no resistive stress $\tau = 0$, it makes sense that surface crack always leads to full-depth penetration (figure 3e1). However, why by increasing the resistive stress to $\tau = 0.04$ (figure 3a1) the surface crack $d_t < 0.15$ would not propagate? This seems to have an important implication, that when the ice surface is fully filled with meltwater, as long as the resistive stress is large enough and the surface crack under certain depth, calving could be induced by basal crack rather than surface crack.

**Response:** We believe that figure 3a1 does show the surface crack $d_t$ propagating for all values $d_t$ above an extremely narrow and visually unresolved strip along the $d_b$ axis[2] The only difference from figure 3e1 is that the basal crack *also* propagates[3] As a result, the orbits shown are no longer vertically but diagonally upwards. No matter where we look in figure 3a1, the orbit are at least a finite angle with the horizontal (if note vertical), indicating that the top crack $d_t$ (plotted along the vertical axis) is growing. It is possible that we have misunderstood the comment, however.

**Reviewer comment:** Second, in the propagation rate model described in equation 4, the crack reaches steady state when the stress intensity factor reaches the fracture toughness. As demonstrated in figure 6, generally when the stress intensity factor curve bends back down at larger crack depth, there exists two steady state crack depths. Are the $K = K_c$ steady states at the smaller $d_t$, $d_b$ simply treated as unstable steady states in the dynamical system approach? Can the author specify this?

**Response:** it's probably not so much that we treat them as unstable steady states as that they are unstable by the usual definition of stability (in the sense that they are *not* Lyapunov stable. We have expanded on this by extending (and splitting) the sixth paragraph of section 3.1 of the original submission, saying

*The usual notions of phase plane analysis, like identifying isolated fixed points and their stability, do not apply without modification due to the non-differentiability of the dynamical system, and due to the fact that equilibria occupy extended regions of the phase plane. Equilibria inside these extended regions are stable in the sense of Lyapunov but not asymptotically stable (Strogatz, 1994): if perturbed, the state variable $(d_b, d_t)$ stays nearby because it does not evolve. For equilibria on the boundary of a region of steady states (that is, equilibria on one of the marginal nullclines), we can distinguish between unstable and stable. The boundary is unstable there are orbits that point away from it, which is the case for boundaries at the top or to the right of a region of steady states, and stable (again in the sense of Lyapunov) otherwise.*

*There are equilibria that occupy a special role, namely those where two marginal nullclines intersect. We will refer to these equilibria as marginal fixed points below. [...]*
* * *
[2]This is again the strip in which the surface crack $d_t$ is too short for the stress intensity factor to rise above $K_{Ic}$, even with the water table at the surface.

[3]This is true at least until $d_t$ reaches some appreciable size, at which point presumably the torques generated by the top crack prevent the bottom crack from growing further.

**Reviewer comment:** Finally, please note that the left hand side of equation 4 is still dimensional; the units on the left-hand side don't seem to align with the unit on the right-hand side. What other system parameters are missing to ensure a matching unit and determine the time scale of crack propagation?

**Response:** The units of the derivative of the "universal function $K$" in the denominator fix thus problem (see Zarrinderakht et al 2022, eqs (18)–(19), or better still, the original derivation in Freund's (1990, chapoter 6) book. The universal function $K$ itself has no units (see equation (6.4.26) and the discussion in the subsequent paragraph of Frenud (1990) as well as equation 6.4.32 of Freund's book for the basis of Zarrinderakht et al's (2022) eq (18)). Notetheless, $K$ is a function of the dimensional velocity $\dot{d}$, so $K'(0)$ has units of one over velocity, ensuring that the right-hand sides of equation (4) have units of velocity. $K'(0)$ scales approximately as $1/(2c_p) - 1/c_R$, where $c_p$ is the primary wave velocity and $c_R$ the Rayleigh wave velocity. We have changed the text around equation (4) to say

*As in Zarrinderakht et al (2022), we assume that each crack propagates at a rate related to how much the stress intensity factor exceeds fracture toughness $K_{Ic}$ by*

$$\dot{d}_{\mathrm{b}} = \max\left(-\frac{K_{\mathrm{Ib}} - K_{\mathrm{Ic}}}{K_{\mathrm{Ic}}|K'(0)|}, 0\right), \qquad \dot{d}_{\mathrm{t}} = \max\left(-\frac{K_{\mathrm{It}} - K_{\mathrm{Ic}}}{K_{\mathrm{Ic}}|K'(0)|}, 0\right),$$

*where the overdot indicates differentiation with respect to time, and $|K'(0)|$ is the derivative of Freund's (1990) universal function $K$ (given by equation (6.4.26) in Freund's book), evaluated at zero crack propagation velocity. An approximate form of the universal function is $K(\dot{d}) \approx (1 - \dot{d}/v_{\mathrm{R}})/\sqrt{1 - \dot{d}/v_{\mathrm{p}}}$, with $v_{\mathrm{R}}$ and $v_{\mathrm{p}}$ being Rayleigh and primary wave velocities, so $-1/K'(0) \approx 2v_{\mathrm{p}}v_{\mathrm{R}}/(2v_{\mathrm{p}} - v_{\mathrm{R}})$. As discussed in ?, there are alternative hydrofracture-based models for crack tip propagation that could replace this description. We pursue the latter here due to the qualitative insights it provides.* We return to the limitations of the propagation model in the updated section 4.2,

*There are several other limitations in addition to not accounting for the effect of buoyancy on elastic stresses. For a given elastic pre-stress, the linear elastic fracture mechanics problem solved here relies on the same weakly inertial propagation rate prescription due to Freund (1990) that was previously used in Zarrinderakht et al (2022). Since the cracks under consideration are typically fluid-filled, it is likely that dynamic propagation is controlled by the retarding effect of fluid flow in the fractures (Spence and Sharpe, 1985), which require a significantly more complicated hydrofracture model, which is unlikely to permit a comprehensive study of parameter space, or even of fracture evolution for different initial conditions as in figures 2, 3 and 5.*

**Reviewer comment:** Eqn 1: Briefly explain the resistive stress Rxx and give a reference

**Response:** We have added the following after equation (1):

*...where $g$ is the acceleration due to gravity, and $R_{xx}$ is related to the far-field velocity field $U$ through $R_{xx} = 4\mu\partial U/\partial x$, $\mu$ being ice viscosity (Muszynski and Birchfield, 1987, Morland 1987, MacAyeal and Barcilon, 1988).*

**Reviewer comment:** Line 115: Add "Poisson's ratio" before $\nu$

**Response:** Changed to

*...is independent of Poisson's ratio $\nu$ (while displacements do depend on $\nu$)*

**Reviewer comment:** Line 117: "written in the form" → "written in the dimensionless form"

**Response:** Changed as suggested.

**Reviewer comment:** Line 122: As surface crack propagates the dimensionless water level generally won't stay constant. Please add a justification or acknowledge the model limitation caused by this simplification.

**Response:** The hydrological assumptions behind this work were discussed in detail in Zarrinderakht et al 2022. There we consider 1) a fixed water table, treating the near-surface of the ice shelf as a porous medium able to support an aquifer and 2) a fixed water volume injected into the crack, one being applicable to warm near-surface conditions and the other to cold surface conditions that still have liquid water available (perhaps seasonally?) We assume the referee would rather use the second version, which is however problematic as explained in Zarrinderakht et al 2022 (in the sense that it is quite possible that fracture propagation in 3D should lead to the formation of localized drainage slots, similar to the instability studied in Touvet et al 2011, see Zarrinderakht et al 2022 for full citation). As for the fixed water table height, assuming that there is abundant near-surface water available in the aquifer, it seems unlikely that the water table would drop appreciably due to the small volume accommodated by a crack in an elastic medium.

To make our assumptions clearer, we have added the following after equation (2)

*Any part of the upper surface below that elevation is also subject to a hydrostatically increasing water pressure, with the water level remaining unchanged as surface cracks propagate. Implicit here is the presence of a near-surface aquifer that can supply sufficient water to fill the crack while maintaining that constant water level.*

**Reviewer comment:** Line 123: "are constant during crack propagation" -¿ "are assumed to be constant during crack propagation."

**Response:** We have reworded this to

*Any changes in forcing parameters are assumed to occur much more slowly than cracks propagate, so the dimensionless forcing and geometry parameters $\tau$, $\eta$, $\kappa$, and $W$ are constant during crack propagation.*

**Reviewer comment:** Line 125: "as t increases" $\rightarrow$ "as time t increases"

**Response:** Changed as suggested.

**Reviewer comment:** Line 134: Have the authors checked that when W further increases the result doesn't change for this aligned surface-basal crack case?

**Response:** We tried a variety values of $W$ including 20, 15, 12, 10, 8 and found no significant difference to our results. This was also done for the results in Zarrinderakht et al (2022), which employ a single crack in the same rectangular geometry. In that case, Tada et al (2000, full reference in the manuscript) provide an interpolated Green's function that was employed in Lai et al (2000, full reference also in the text) and we checked for agreement with that.

**Reviewer comment:** Line 141: "maximum ensures not only ensures that cracks cannot shrink" The first "ensure" appear to be a typo

**Response:** Indeed.

**Reviewer comment:** Line 142: What variable is non-differentiable against what variable? $K$ against $d_b$, $d_t$?

**Response:** Both $K$'s may well be differentiable with respect to $d_\mathrm{t}$ and $d_\mathrm{b}$. That does not mean the right-hand side of equations (7) is differentiable: this is basically a slightly more complicated manifestation of the fact that $f(x) = x$ is differentiable with respect to $x$, but $g(x) = \max(x, 0)$ is not.

To make this clearer, we have recorded the section slightly to say:

*Second, the dynamical system is non-smooth: the maximum function on the right-hand sides of equations (7) not only ensures that cracks cannot shrink, it generally renders those right-hand sides non-differentiable where $K_\mathrm{Ib}^* = \kappa$ or $K_\mathrm{Ib}^* = \kappa$, even if $K_\mathrm{Ib}$ and $K_\mathrm{It}$ are smooth functions of $(d_\mathrm{b}, d_\mathrm{t})$ (where the latter seems likely unless a new contact area is formed, or a section of open crack fully disappears at that point, see figure 4a of Zarrinderakht et al (2022)).*

In other words, we explicitly state that we are concerned with the differnetiability of the right-hand

sides of equations (7), not of the $K$'s, and that we actually expect the $K$'s to be differentiable (which we cannot prove). The referenced figure illustrates the non-differentiability for the case of a single crack.

**Reviewer comment:** Line 144: "intensity factors is equal to the fracture toughness." -¿ "intensity factors is equal to the fracture toughness (green and yellow lines in figure 2)."

**Response:** That is not strictly speaking true. The original text did not define all of these coloured curves, but they are in fact the orbits into and out of the green-marked saddle point, and the orbits that delimit the basin of attraction of the red-marked node point. They are *nearly* horizontal and vertical for the most part, but not exactly so, as they would need to be if they did coincide with the marginal nullclines. This is probably most evident for the green orbit that emerges near-vertically from the saddle point, which curves noticeably to the right near the calving boundary.

We have changed the text to clarify what the coloured orbits are. In the figure caption for figure 2, *The coloured dots indicate saddle-type (green, yellow and magenta) and node-type (red and cyan) marginal fixed points as defined in the main text. The green curves are the orbits into and out of the green saddle point, while the red and yellow curves are the orbits that delimit the basin of attraction of the red node point.* and in the main text for the red and yellow curves (note that one of the red curves of the original manuscript has been removed as it played no special role, but the yellow orbits of the inset have been extended and added to the main figure panel

*Even though the stable node is not an attractor in the strict sense (there are other equilibria arbitrarily close to the stable node), it does have a finite basin of attraction demarcated by the red and yellow orbits into the stable node. Note that he size of that basin of attraction is easy to overestimate visually due to the finite resolution used in computing the phase portrait. Close to the stable node is a marginal fixed point that is analogous to a saddle in standard phase plane analysis, marked with a yellow dot in the inset. For this marginal fixed point, a single orbit ends at the saddle, while a second orbit connects saddle and node (both shown in yellow in the inset). Below the orbit leading up to the saddle, there are additional orbits starting with lower values of $d_t(0)/d_b(0)$ that terminate at the boundary of a region of steady states as shown in the inset.* and for the green curves *Here, one orbit (marked in green) emerges from the saddle point towards the calving boundary while a second orbit (also marked in green) connects a fourth (cyan) marginal fixed point that is almost on the $d_t$-axis to the green saddle point; this orbit is analogous to a separatrix in a standard phase plane, and divides initial conditions that lead to immediate calving from initial conditions that lead to stable, steady cracks of finite length that leave the ice slab intact.*

**Reviewer comment:** Figure 7: Is the critical stress to drive calving sensitive to the resolution of your numerical model? If yes. Have the authors checked that the result had already converged with higher resolution?

**Response:** We tested our results by halving and doubling resolution, and found no noticeable change.

**Reviewer comment:** line 385: "We anticipate that incorporating the feedback between displacement and fluid pressure at the boundary will lead to additional torques generated by vertical displacements in the far field, suppressing crack growth for very large crack spacings". The effect of vertical elastic deformation on buoyancy at the ice-ocean interface was included in Buck and Lai (2021), although their result corresponds to zero fracture toughness. The ice-ocean restoring buoyancy force can increase the critical stress $\tau_{crit}$ for basal crack to reach the sea level.

**Response:** Indeed; this was a restatement from Zarrinderakht et al 2022. Apologies for having dropped the reference to Buck and Lai (2021), which has been restored.

---

## Author Comment (AC2)

**Response to RC2: 'Comment on egusphere-2023-1252', Jeremy Bassis, 25 Sep 2023**

January 30, 2024

**Reviewer Comment:** This study is the second (or third) in a trilogy of papers by the same author team that examines propagation of crevasses in freely floating ice shelves using a boundary element model. The contribution of this manuscript is to study the effect of surface water filled and longitudinal extension on the interaction between surface and bottom crevasses.It has long been assumed that surface and basal crevasses intersect to form rifts, but the dynamics of surface and basal crevasse propagation and interaction have rarely been studied. Hence, this is a welcome study into an old, but important problem.

The problem of the interaction between adjacent cracks also has a long history in the fracture mechanics literature. This is a surprisingly challenging problem because, even under pure model I loading, the crack tip stress field from interaction results in mixed-mode loading. As a consequence, experiments and theory indicate that en echelon cracks coalesce in a "kink" rather than in a straight intersection. The analytic, numerical and experimental results that I am familiar with for this problem are typically done under idealized pure mode I propagation so it isn't clear that this necessarily applies to the ice shelf problem. Nonetheless, it would be reassuring if the authors can use their model to reproduce some of the classic results of en en-echelon elastic fracture propagation—or at least touch base—with some of the literature. I would be surprised if the interaction between surface and bottom crevasses resulted in pure mode-I behavior. The fact that this problem has a lot of history, in my opinion, deserves a little bit more attention in the introduction.

**Response:** Thank you for bringing this up as something to be flagged early in the paper. The method that we are using is capable of determining the direction of crack propagation through a maximum hoop stress or maximum energy release criterion (which we plan to implement in this context in the next iteration of the code). Here we have, in a sense, "cheated" to create or model set-up where crack orientation is prescribed, ot facilitate what we see as the simplest setting for interacting cracks (in which we can really take a dynamical systems approach of representing cracks geometry by crack length alone!). "Cheated" however only in very limited ways: the model we use is perfectly self-consistent in assuming crack orientation is fixed because the stress field is guaranteed to be of mode I type through symmetry. This should be obvious for the aligned cracks in figure 1. For the offset cracks of figure 4, it is important to recall that the domain is periodic, which ensures that the stress field possesses the necessary reflection symmetry about either crack. This differs from en echelon cracks, where such symmetry does not exist regardless of the far field forcing.

The cheat in our manuscript is two-fold in that sense. In reality, cracks can form anywhere and we should be seeding the domain with lots of incipient cracks that can grow and change direction. (As per the above, that is next on the docket for this work.) The second, and perhaps more relevant cheat is that we assume that the cracks do not branch or develop kinks through an instability, where even though the stress field is symmetric about the crack tip, the greatest hoop stress could be offaxis and attract or repel the cracks from each other. We discuss these issues (without pretending to be providing a comprehensive review):

*To keep the scope of our work tractable, we restrict ourselves to understanding simple interactions between basal and surface crevasses. In particular, we seek to identify how the spacing and alignment of crevasses on opposite sides of an ice shelf affect calving. Note that the study of interacting cracks has a long history, often involving complicated geometries in wich the direction of crack propagation must be determined as part of the solution of the linear elastic fracture mechanics problem (e.g. Seagall and Pollard, 1980; Baud and Reuschlé 1997). Here we use the fact that two-dimensional ice shelves flow in pure shear at leading order (Morland, 1987) to restrict ourselves to simple crack geometries in which the stress field remains symmetric about each crack and the assumption of vertical crack propagation remains self-consistent with a maximum hoop stress criterion Zehnder (2010).*

In the second paragraph of section 3.3, we have also added

*Note that the assumption of vertical crack propagation is then consistent with a maximum hoop stress criterion (Zehnder 2010, setion 4.4.1) but we do not address the question of crack path stability (Cotterell and Rice 1980), namely that a perturbed crack could evolve progressively away from a vertical orientation.*

In addition, we reiterate the point in the penultimate paragraph of section 4.2, ... *In addition, the assumption of purely vertical crack propagation is contingent on the highly specific crack orientations considered here, which ensure that we have purely mode 1 crack propagation. In reality, there are likely to be many interacting and potentially curved cracks, which we will address with a future iteration of the model.*

**Reviewer comment:** I mentioned this in my previous review of a different manuscript, but I find the non-dimensionalization counter-intuitive and hard to track. The two main dimensionless numbers are $\tau$ and $\eta$. The parameter $\tau$ is a measure of longitudinal extension and eta is a measure of the water pressure filling crevasses. A more natural (to me) definition of tau would define the non-dimensional longitudinal extension stress based on the reduced gravitational acceleration ($g' = (1 - r)g$) or, equivalently, based on the resistive stress associated with a freely spreading ice shelf. This would imply that a value near unity corresponds to an ice shelf spreading under its own weight and values larger or smaller would correspond to extensional stresses that are larger or smaller compared to an ice shelf spreading purely under its own weight. I have a hard time visualizing what a $\tau$ of 0.02 means physically without resorting to using my calculator to mess with densities. I think the $\eta$ parameter is even more difficult for me to visualize. The situation most relevant for most ice tongues is the surface-water free case. Previous studies have defined water depth in crevasses as a fraction of the crevasse depth, which is a bit more intuitive to visualize (brim-full vs empty). I would encourage the authors to consider their non-dimensionalization and to connect the values as much to physical situations as possible (i.e., water-free crevasses, extension larger/smaller than the gravitationally induced spreading, etc.) to make it as easy as possible for readers to understand the underlying physical situation the authors envision.

**Response:** Thank you for pointing this out — having settled on a notation, it is easy to start imagining that that notation is "natural".

The suggestion of scaling using reduced gravity is attractive in principle, certainly for the simple underlying parallel-sided slab geometry in the present paper. Presumably, this would be $\tau = R_{xx}/(\rho_i g' H)$ as the simpler version, or $\tau = 2R_{xx}/(\rho_i g' H)$ if you wanted to have $\tau = 1$ for an unconfined 2-D ice shelf, the factor of 2 being the result of the usual depth-averaging

The reason why we did not use a reduced gravity variable for the problem is the Stokes flow problem for the viscous pre-stress studied in the companion paper referenced in the comment above. If we did

use a reduced gravity variable, that naturally would go with a reduced pressure $p' = p + \rho_i gz$. The reduced gravity scaling and reduced pressure variable leads to a simpler (body-force-free) version of the Stokes equations and simplifies the boundary conditions for the part of the lower boundary below the water line. The cost is that the boundary conditions at the upper surface become more complicated (a moot point for most buoyant fluid flow problems, which replace that upper boundary with a rigid one), and the density ratio $r$ actually cannot be scaled out of the problem — making the scaling used here seem like the better choice at the time of writing.

That said, we are still hesitant to change the scaling here, because (at least for $\tau$), it is the same scaling used previously in Lai et al (2020) and Zarrinderakht et al (2021), and it seems unwise to make a reader who is reading these papers in sequence translate from one scaling to another. (Notably the scaling for $\tau$ is also used in a number of theory papers on marine ice sheets / tidewater outlets by one of us (CS) as well as others (Sergienko and Haseloff, primarily).

We're hoping that this can be resolved by explaining better how to read the numerical values of $\tau$, which differ from the ones you would get with the reduced gravity scaling above by a factor of $g'/g = (1 - r) \approx 0.11$. We have reworded the ninth paragraph of section 2 (which introduces the non-dimensionalization) as follows, providing typical values of $\tau$ and $\eta$ to guide the reader:

*To simplify the set of geometrical and forcing parameters, we non-dimensionalize the model using the same set of scales as in Zarrinderakht et al (2022) and Lai et al (2020). This leaves only the following dimensionless parameters,*

$$\tau = \frac{R_{xx}}{\rho_i gH}, \qquad \eta = \frac{h_w}{H}, \qquad \kappa = \frac{K_{Ic}}{\rho_i gH^{3/2}}, \qquad W^* = \frac{W}{H}, \tag{1}$$

*in addition to the dimensionless material constants given by Poisson's ration $\nu$, and*

$$r = \frac{\rho_i}{\rho_w}. \tag{2}$$

*Above, $\tau$ is a dimensionless extensional stress, $\eta$ a dimensionless depth to the surface water table, and $\kappa$ a dimensionless fracture toughness. We will primarily focus on dimensionless extensional stress $\tau$ and water level $\eta$ as forcing parameters, since $\kappa$ is likely small: with a dimesional fracture toughness $K_{Ic} = 0,4$ MPa $m^{-1/2}$ (Rist et al, 1996) and an ice thickness of $H = 500$ m, $\kappa \approx 0.004$. To understand better how to map the dimensionless parameters to dimensional ones, recall that the extensional stress in an unconfined, one-dimensional ice shelf is $\rho_i(1 - r)gH/2$ (van der Veen, 1983; MacAyeal and Barcilon 1988). With a density ratio of $r = 0.89$, this corresponds to $\tau = 0.055$, which provides a reference value for the dimensionless extensional stress. The water level parameter is somewhat simpler: $\eta = 0$ corresponds to completely full surface cracks with the water level at the upper surface. $\eta = 1$ corresponds to a surface crack that remains dry no matter how far it is incised. A value of $\eta = 1 - r = 0.11$ represents a surface crack for which any portion below sea level is filled with water.*

At risk of sounding patronizing, the meaning of $\eta$ should be easier to deal with once explained, as it ranges from $\eta = 0$ when surface crevasses are "brim-full" to $\eta = 1$ for surface crevasses that remain empty no matter how deep they are, with $\eta = 1 - r$ holding additional significance by representing surface crevasses that start to fill with water when their tip reaches sea level. The proposed change in the text above should cover this along with the interpretation of $\tau$.

The comment includes an additional suggestion / reference to modelling crevasses as being filled to a certain fraction of their length. That would be more than a change in the non-dimensionalization, but represent an entirely different way of forcing surface crevasses. We struggle, however, to envision a surface hydrology that would lead to this outcome — of the water level simply dropping in

proportion to the length of the crevasse. Our earlier paper (Zarrinderakht et al 2020) considers to hydrology end-member that we consider plausible, namely (1) a fixed surface water level fed by some form of aquifer that acts as a buffer to water level changes as the crevasse propagates and widens and (2) a fixed volume of water injected into the crevasse, the latter choice turning out to be somewhat problematic for the purposes of modelling calving

**Reviewer comment:** One of the novelties of this study is the display of basal and surface crevasse depths in a phase plane. I think this is an interesting way of displaying the results with a lot of potential. This method introduces a slightly different perspective than the way we typically think of these problems. The way we normally think of the system is how deep will a crevasse penetrate given a small "starter crack" of some pre-determined size. The phase plane encourages us to think about pre-existing crevasses of a variety of sizes, including those that aren't necessarily "small". The question that this introduces is what processes introduce large-is crevasses that seed the initial conditions? Is the idea that crevasses advect from a region where the stress was larger? I see the authors come back to this on page 15. It might be helpful to foreshadow or mention this earlier. This is especially relevant because what we typically see is that rifts and crevasses initiate along the margins and propagate from the margins into the interior of the ice shelf. This requires a more 3D treatment of fracture, but it seems relevant that the starting depth for basal or surface crevasses here might be related to the horizontal propagation of a crevasse or rift with some stress that includes stress concentrations associated with the horizontal fracture.

**Response:** We admit to being daunted by the prospect of doing this in three dimensions. The point is however well made: why would you consider any initial conditions other than a small seed crack? We telegraph the later development of this idea at the end of section 2,

*The ability to visualize evolution from arbitrary initial conditions using a phase plane also allows us to address how the dynamical system evolves under slow changes in forcing parameters (see also Zarrinderakht et al, 2022, sections 4-4–4.5): if started with a combination of forcing parameters that does not cause calving (generally with $\tau$ being too small or $\eta$ too large), partially incised crevasse will still typically result. A subsequent change in parameters may then lead to full crack penetration starting with initial conditions dictated by the previous formation of a partially incised crack (as opposed to short seed cracks only), subject to the caveat that we do not re-compute the full viscous pre-stress in this paper when doing so (but see also Zarrinderakht et al, submitted).*

**Reviewer comment:** I think this might be addressed in one of the other manuscripts, but when the authors introduce a crevasse into a freely floating ice shelf, the ice shelf has a flexural response that is not incorporated by the "viscous pre-stress". The flexural response tends to reduce the stress concentrated ahead of crevasses. Is this included in the boundary element model? What effect would neglecting it have on model results? What does the flexural stress do to the lateral boundary conditions? I assume this is negligible for domains that are very large compared to the flexural wavelength, but the domain sizes here seem roughly comparable to the flexural wavelength or smaller. This seems especially relevant to the interaction between crevasses. I see this is returned to near lines 385. I think it might be worth introducing this earlier, perhaps in the methods/model section as it seems quite important.

**Response:** This depends on a bit on what flexure means in context. If we are talking about bending moments induced by transverse (vertical) displacements, then the model does account for those: the model is the full elastostatic version of Navier-Cauchy equations for plane strain, for which the behaviour of an elastic beam is the appropriate long-wavelength (or far field) behaviour. The boundary element aspect is simply the method by which the model is solved in discretized form; the same problem could be solved with for instance an XFEM solver, or an FEM solver and some suitable method of computing a $J$-integral.

Beam-like behaviour should be evident in e.g. figure 8b, especially if you imagine that extended periodically to the left and right. What the model does *not* include is the buoyant restoring force that results from flexural uplift in the far field. In the near-field (over horizontal distances comparable with ice thickness scale $[H]$), that neglect is appropriate in the small strain limit that underpins the rheology: uplift is so small that hydrostatic changes in fluid pressure at the boundary have to remain small (the error in omitting them being comparable to the strain). That fails in the far field, at horizontal distances comparable to $\{E/(rho_\mathrm{i}g[H])\}^{1/4}[H] = [\varepsilon]^{-1/4}[H]$, $[\varepsilon]$ being the scale for strain, which is presumably the flexural wavelength in question. Here, vertical displacements are large enough that they affect the stress field at leading order through buoyancy effects. This is discussed in Zarrinderakht et al (2022) at the top of page 4495 (note that there is a typo in the definition of the flexural length scale on line 10 of that page (the exponent should be $-1/4$ rather than $1/4$). Implications are discussed in more detail in section 6.3 of Zarrinderakht et al (2022; see especially figure 10 therein), where we point out that a model that neglects the buoyant restoring force most likely underestimates the critical extensional stress at which calving due to basal crevasse propagation occurs (as the original comment indicates, flexure will reduces stresses around the crack).

The same point as in section 6.3 of Zarrinderakht et al (2022) is reiterated in section 3.4 of the present manuscript (around line 385 in the original submission as identified in the reviewer comment) and again in the second paragraph of section 4.2,

*We anticipate that incorporating the feedback between displacement and fluid pressure at the boundary will lead to additional torques generated by vertical displacements in the far field, suppressing crack growth for very large crack spacings. We leave a study of this effect to future work.* We have added a note to the effect that buoyancy effects are neglected in the model in the updated section 2, appending the following to the second paragraph:

*As in Zarrinderakht et al (2022), we ignore the effect of elastic displacements on the fluid pressure at the boundary, thereby omitting buoyancy effects. This is a potentially significant omission that affects large-scale flexure effects as discussed further in section 4.2 below (see also sections 2.1 and 6.3 of Zarrinderakht et al (2022)).*

Beyond that, we plan to incorporate buoyancy effects in the next iteration of the model, and would probably prefer to deal with the issue in detail then, rather than speculating further here.

**Reviewer comment:** Is it true that vertical propagation is the most optimal orientation for crevasse propagation? If the direction of propagation is determined by the direction of maximum principal stress, are crevasses expected to kink or turn based on the direction of maximum principal stress? A relevant physical question is what happens to crevasses that are slightly offset from each other? It would be surprising if crevasses were exactly aligned, but what if they are mis-aligned by a small fraction of the width? Would they never intersect? Is it possible that the phase space is not well resolved if crevasses are allowed to kink or turn?

**Response:** This is probably covered in the response to the first major comment of the review, see above. The crevasse orientation and alignment / spacing is chosen to make sure that vertical crack propagation in pure mode 1 is self-consistent, but that does not ensure that that orientation is stable to turning / kinking, as the paper now states explicitly.

**Reviewer comment:** Line 18-35. I think the more relevant comparison is between boundary element models and damage mechanics. Damage mechanics can be used so simulate failure under a wide variety of circumstances. Judicious choice of the damage production function allows damage mechanics to reproduce LEFM results, creep rupture or any heuristic method of simulating failure. One of the reasons that damage mechanics is so popular is that it avoid the need to remesh that is the bane of many LEFM simulations. Damage mechanics has been used to simulate the growth

of both isolated surface and basal crevasses and arrays of crevasses. It would be nice to a more detailed comparison between the results considered here and those previous results.

**Response:** Our original statement here was misleading — it was mostly intended to refer to the observation that, if you "refine" a discrete element network by for instance halving the spacing $d$ between discrete elements, then the breaking strength (that is, the size of hte force required to break a bond) presumably does not simply scale with element spacing: near a crack tip, a continuum model would predict that stress satisfies a one-over-square-root relationship with distance from a crack tip, and therefore bond forces near crack tips should scale as $d^{1/2}$, rather than with $d$. Bond strength should therefore scale as $d^{1/2}$ (rather than $d$?), and it was not clear to us that this is what is usually done in discrete element models — LEFM provides a systematic way around this by computing $K_I$ (with corresponding methods that converge under mesh refinement).

Anyhow, we have removed the original discussion of discrete elements in favour of the text below, and have added additional text about phase field methods (which do agree well with LEFM — as they are intended to — albeit at additional computational cost (offset by their flexibility in capturing crack initiation and complex crack geometries). More generic damage mechanics methods do seem harder to reconcile with LEFM as the set-up us fundamentally different, and the damage production parameter (usually $\hat{B}$) is independent of other model parameters but must play a very important role, since it determines whether damage evolves significantly faster than viscoelastic stress relaxation and attendant crack tip blunting etc:

*Discrete element models (Bassis, 2011, Åström et al 2013, Crawford et al 2021) are better able to cope with multiple interacting cracks, and with cracks of arbitrary geometry, but they are computationally expensive and therefore difficult to apply when exploring larger regions of parameter space. More recently, phase-field models for fracture mechanics have been applied to crevasse formation (e.g., Clayton et al 2022, Sondershaus et al, 2023), which reproduce the predictions of linear elastic fracture mechanics closely while also being able to handle phenomena such as crack splitting and viscoelastic relaxation of stresses (though, at present, seemingly only for small viscous strains). As with discrete elements, phase field models however are also computationally more expensive than classical linear elastic fracture mechanics approaches., requiring additional degrees of freedom to be solved for. Note that more general damage mechanics models (Duddu et al, 2014, Duddu et al, 2020, Jiménez et al 2017, Keller and Hutter 2014, Mobasher et al 2016) aim in a similar direction, but unlike phase field models are not ostensibly based on the energetics of creating new fracture surfaces, and introduces additional parameters that control not only a critical stress for damage production, but also the rate of damage production, which makes comparison with models based on fracture mechanics more difficult.*

**Reviewer comment:** Line 33. It is true that discrete element models do have a dependency on the packing orientation, but it has been shown that these models to converge to the continuum elastic limit under some circumstances. One of the open questions, however, is how to specify the bond strength. Conventional discrete element models include two fracture parameters and this allows mixed-mode failure. Mixed-mode failure is something that can also be difficult to simulate within a linear elastic fracture mechanics framework because it requires an additional criterion to allow cracks to kink or bend. Typically, one assumes that cracks propagate in the direction of the largest principal stress. It seems like this study, however, assumes single mode loading.

**Response:** We hope this is already covered by the response to the previous comment and the response to the first comment of the review. The boundary element method used here can be adapted to curving cracks (by something analogous to what is done in

E Gordelyi, S Abbas and A Peirce (2019) Modeling nonplanar hydraulic fracture propagation using the XFEM: An implicit level-set algorithm and fracture tip asymptotics, Int. J. Solids and Structures, 159, 135–155

and that is indeed what we plan to do next. The short answer is however "yes", the current paper imposes boundary conditions (through symmetry in a periodic domain or otherwise) that ensure single mode fracture propagation.

**Reviewer comment:** Line 135: Vanishing elastic traction implies that elastic strains vanish at the domain boundaries, but least displacements are allowed, right?

**Response:** Yes. We don't think that you can impose both vanishing strain and displacement. You could have mixed conditions (one component of traction and one component of displacement vanishing) but not both components of traction and displacement vanishing, since that would be an overdetermined elliptic system.

**Reviewer comment:** Equation (4). What are the units and numerical value of K'(0)? I apologize if I missed this in the manuscript. If K'(0) is dimensionless, it is unclear how the units of the equation work out. If it is dimensional, then we need to know the numerical value.

**Response:** This wasn't particularly well described in the original submission, with just a blanket reference to Freund's book on dynamic fracture propagation. We have changed the text around equation (4) to say

*As in Zarrinderakht et al (2022), we assume that each crack propagates at a rate related to how much the stress intensity factor exceeds fracture toughness $K_{Ic}$ by*

$$\dot{d}_{\mathrm{b}} = \max\left(-\frac{K_{\mathrm{Ib}} - K_{\mathrm{Ic}}}{K_{\mathrm{Ic}}|K'(0)|}, 0\right), \qquad \dot{d}_{\mathrm{t}} = \max\left(-\frac{K_{\mathrm{It}} - K_{\mathrm{Ic}}}{K_{\mathrm{Ic}}|K'(0)|}, 0\right),$$

*where the overdot indicates differentiation with respect to time, and $|K'(0)|$ is the derivative of Freund's (1990) universal function K (given by equation (6.4.26) in Freund's book), evaluated at zero crack propagation velocity. An approximate form of the universal function is $K(\dot{d}) \approx (1 - \dot{d}/v_{\mathrm{R}})/\sqrt{1 - \dot{d}/v_{\mathrm{p}}}$, with $v_{\mathrm{R}}$ and $v_{\mathrm{p}}$ being Rayleigh and primary wave velocities, so $-1/K'(0) \approx 2v_{\mathrm{p}}v_{\mathrm{R}}/(2v_{\mathrm{p}} - v_{\mathrm{R}})$. As discussed in ?, there are alternative hydrofracture-based models for crack tip propagation that could replace this description. We pursue the latter here due to the qualitative insights it provides.*

**Reviewer comment:** Line 245: Placing cracks a distance of W/4 and 3W/4 depends on the width of the domain. What about slightly offset crevasses? There is, in theory, two distances in the problem, right? The distance between the crevasses and the length of the (periodic) domain. What happens if the distance between crevasses remains the same, but the length of the domain increases?

**Response:** That is the point at which the symmetry conditions that lead to single mode fracture propagation fail. We hope the response to the first comment of the review (and attendant changes to the manuscript) cover this adequately.

**Reviewer comment:** Line 87: Punctuation? Is the semi colon supposed to be there?

**Response:** Oops. That should have been a full stop, and the "to" has no place here either. Corrected to say

*The symmetry conditions we impose on their locations below makes that choice of orientation self-consistent.*

**Reviewer comment:** I think lines 295 are saying that you need a larger stress to propagate an array of crevasses all the way through compared to isolated crevasses. This is consistent with previous analytic calculations by Weertman and others.

**Response:** This is true, in a slightly subtle sense: Weertman (1973) and van der Veen (1998a) (the two studies in which we are aware of this result) deal with crevasses in an infinitely deep ice domain

when looking at interacting crevasses, and find that the relationship between penetration depth to spacing controls how much $K_I$ is reduced relative to the case of an isolated crevasse subjected to the same spacing. The limitation to an infinite depth corresponds to a crevasse that has penetrated only a small fraction of the full ice thickness, and a lateral crevasse spacing comparable such a small depth would be the limit of a small crevasse spacing in our model (distances being scaled with ice thickness $H$). We have added to the text here to say

*Note that in the limit of a small crevasse spacing (much less than a single ice thickness), the effect of neighbouring crevasses observed here agrees with the previous results of ? and ?, who found a significant reduction in crack tip stress intensity factor for crevasses that are spaced closer than their depth of penetration, relative to an isolated crevasse.*

---

## Author Response (AR1)

**Public justification (visible to the public if the article is accepted and published)**:
Dear Maryam and co-authors,

Thank you for your detailed response to reviewer's comments. I now invite you to submit the revised version of your manuscript which will then be considered by myself and the reviewers.

Best wishes,

Caroline

Thank you, Done!

---

## Author Response (AR2)

**Dear Caroline Clason,**

A revised version of our manuscript considering the referee's comments is prepared.

Best wishes,

Maryam Zarrinderakht, Christian Schoof and Anthony Peirce